# Presentation and evaluation of the Arctic sea ice forecasting system neXtSIM-F

Timothy Williams[1], Anton Korosov[1], Pierre Rampal[2,1], and Einar Ólason[1]

[1]Nansen Environmental and Remote Sensing Center, Thormøhlensgate 47, 5006 Bergen, Norway and the Bjerknes Center for Climate Research, Bergen, Norway
[2]Université Grenoble Alpes/CNRS/IRD/G-INP, Institut Géophysique de l'Environnement, Grenoble, France

**Correspondence:** Timothy Williams (timothy.williams@nersc.no)

**Abstract.** The neXtSIM-F forecasting system consists of a stand-alone sea ice model, neXtSIM, forced by the TOPAZ ocean forecast and the ECMWF atmospheric forecast, combined with daily data assimilation of sea ice concentration. It uses the novel Brittle Bingham-Maxwell (BBM) sea ice rheology, making it the first forecast based on a continuum model not to use the viscous-plastic (VP) rheology. It was tested in the Arctic for the time period November 2018 – June 2020 and was found to perform well, although there are some shortcomings. Despite drift not being assimilated in our system, the sea ice drift is good throughout the year, being relatively unbiased, even for longer lead times like 5 days. The RMSE in speed and the total RMSE are also good for the first 3 or so days, although they both increase steadily with lead time. The thickness distribution is relatively good, although there are some regions that experience excessive thickening with negative implications for the summer-time sea ice extent, particularly in the Greenland Sea.

The neXtSIM-F forecasting system assimilates OSI-SAF sea ice concentration products (both SSMI and AMSR2) by modifying the initial conditions daily and adding a compensating heat flux to prevent removed ice growing back too quickly. The assimilation greatly improves the sea ice extent for the forecast duration.

## 1 Introduction

Arctic sea ice has been in great decline in the last number of years (Meier, 2017). Perovich et al. (2018) report that in 2018, the summer extent was the sixth lowest and the winter extent was the second lowest in the satellite record (1979–2018). Moreover, surface air temperatures in the Arctic continued to warm at twice the rate relative to the rest of the globe, and Arctic air temperatures for the past five years (2014-18) have exceeded all previous records since 1900 (Overland et al., 2018), which will also contribute to future sea ice decline if it continues.

With less sea ice comes an increase in summertime accessibility for shipping. Azzara et al. (2015) considered a range of different scenarios and projected an increase in the number of vessels operating in the Bering Strait and the U.S. Arctic of between 100 and 500%. The International Maritime Organization has also recognized that shipping would increase and adopted an International Code for Ships Operating in Polar Waters (Polar Code)[1] on 1 January 2017. This polar code addresses the

---

[1]http://www.imo.org/en/MediaCentre/HotTopics/polar/Pages/default.aspx

increased safety and pollution risks of operating in the Arctic. A recent example of the risks and concomitant costs of accidents in the Arctic is the rescue of the fishing vessel Northguider, which ran aground between Spitzbergen and Nordaustlandet (Svalbard) after getting into trouble with sea ice. The crew had to be rescued by the Norwegian Coast Guard icebreaker K.V. Svalbard, who then had to drain 300 kl of diesel from the damaged vessel.[2]

Thus sea ice forecasting is becoming increasingly important. As well as search and rescue/accident prevention, other applications are optimized ship (icebreaker) routing based on forecasts (Kaleschke et al., 2016) and support of research activities – e.g. Schweiger and Zhang (2015) give an example of scheduling of high-resolution SAR images in order to follow the drift of some ice-mass balance (IMB) buoys by using the PIOMAS/MIZMAS forecast from the University of Washington. The year-long drift of the Polarstern from September 2019 (part of the MOSAiC project) also relied heavily on sea ice and weather

forecasts.

    Tonani et al. (2015) give a good overview of the 2015 status of operational forecasting (here we take "operational forecasts" to refer to those with forecast horizons of about a week), while Hunke et al. (2020, Table 1) give many examples of modelling systems that include sea ice, most of which are used operationally in national forecasting capacities. They vary in resolution, in complexity (with regards to the modelled processes and the coupling between these processes) and in the data assimilation

schemes that are used. We note however that their sea-ice dynamics schemes are all based on Eulerian advection schemes and on variants of the viscous-plastic (VP) rheology (although some solve the rheological equations directly while others solve modified equations as is done with the elasto-viscous plastic method (EVP)). In this review, Hunke et al. (2020) note that the trend is towards fully-coupled systems a high resolution. For example, the ECMWF forecast coupled ice and ocean models to their atmospheric model in between the two papers (this system went operational in June 2018) at a resolution of $0.1°$.

neXtSIM-F uses this latest ECMWF product (IFS: integrated forecast system: Owens and Hewson, 2018) to provide forecast atmospheric forcing, along with ocean forcing from TOPAZ (Sakov et al., 2012).

    Another relevant example is the replacement of RIPS[3] (Lemieux et al., 2016a) by RIOPS[4] in 2016, having NEMO coupled to the system (Smith et al., 2021). The move was partly motivated by wishing to have detailed currents forecast around the Canadian coast for search-and-rescue purposes, including tidal forecasts. RIPS used a stand-alone sea ice model based on

the CICE sea ice model which used 3DVAR assimilation of concentration retrievals from passive microwave (Special Sensor Microwave Imager (SSMI) and Special Sensor Microwave Imager/Sounder (SSMIS)), advanced scatterometer data, and ice charts from the Canadian Ice Service.

    In this paper, we introduce a new sea ice forecasting system, neXtSIM-F, which is based on a stand-alone version of the sea ice model neXtSIM (Rampal et al., 2016, 2019). Not being part of a coupled system it is quite dependent on the atmospheric

and oceanic forcings, which are quite influential on things like the ice edge location and how long corrections to the ice edge persist if the ice edge in the forcings are incorrect. On the other hand, not having an ocean model, the computational cost of the

---

[2]https://www.highnorthnews.com/en/svalbard-preparing-extreme-pumping-operation-using-small-boats

[3]Regional Ice Prediction Service: operated by ECCC (Environment and Climate Change Canada) from 2013 to 2016.

[4]Regional Ice Ocean Prediction System: operated by ECCC. See https://eccc-msc.github.io/open-data/msc-data/nwp_riops/readme_riops_en/#technical-documentation.

system is quite low, and it is relatively stable to perturbations during assimilation — something which gives a certain amount of flexibility to the possible approaches to assimilation.

neXtSIM is a Lagrangian finite element model, and we are running it with a nominal triangle side length of 10 km, with a distance from one point of a triangle to the opposite edge being about 7.5 km. The name neXtSIM-F refers to the entire platform, including data input/output and assimilation, model initialisation and simulation, export, visualisation and evaluation of results (see Section 3). Due to the relatively recent arrival of the sea ice model, neXtSIM-F is simpler than most other platforms, both in terms of assimilation scheme (data insertion with nudging) and model components (uncoupled to ocean or atmosphere). (See Section 5 for planned improvements.) However, it is the first forecasting system based on a model with brittle sea ice rheology instead of the traditional viscous-plastic (VP) family of rheologies. It is also the first system being based on a Lagrangian (adaptive) deforming grid, as opposed to the other ones being based on the standard Eulerian (fixed) grids.

neXtSIM-F entered into operations as part of the Copernicus Marine Environment Monitoring Services (CMEMS) on 7 July 2020. It was equipped with neXtSIM v1.0 based on the Maxwell-Elasto-Brittle (MEB) rheology (Dansereau et al., 2016), which had been shown to reproduce Arctic sea ice drift and deformation particularly well (Rampal et al., 2016, 2019). MEB consisted of an elastic spring in series with a dashpot, together with two main modifications to improve localisation and to prevent excessive convergence: the damage value was only used to modify the stress when it exceeded a threshold of 0.95, and a kind of viscoplastic stress term was added which only played a role when the ice was very damaged. This improved the thickness field somewhat over longer simulations (1–2 years: Rampal et al., 2019), but not quite enough for longer than that.

In September 2020 the core of the forecasting platform was replaced with a new model: neXtSIM v2.0 based on a preliminary version of the novel Brittle-Bingham-Maxwell (BBM) sea ice rheology (Ólason et al., in prep.). This newer version of neXtSIM-F entered into operations in December 2020.

BBM consists of an elastic spring in series with a composite element that contains a dashpot and a frictional sliding element in parallel (Ólason et al., in prep.). (For a summary of the BBM equations, see Appendix A.) The BBM's main physical achievement has been to stabilise the thickness for decadal-scale simulations; computationally it is also 5–6 times faster, since it is able to be solved explicitly, unlike our version of the MEB. It also kept the improvements in the model's representation of the main spatial and temporal characteristics of observed deformation that were gained by MEB: strain localization and scaling (Marsan et al., 2004; Rampal et al., 2008; Stern and Lindsay, 2009), multifractality and intermittency in time (Rampal et al., 2019), meaning that higher deformations are more localised in space and more intermittent in time than smaller ones. These properties have strong implications for things like distribution and size of lead openings and how long they will stay open, which controls the heat and salt fluxes across the ocean–ice–atmosphere coupled system. While the precise forecast of individual leads is very challenging, and probably requiring assimilation of quite specific data like SAR-derived deformation (Korosov and Rampal, 2017), reliable information of this sort would be very useful for icebreakers that wish to reduce fuel consumption or submarines wishing to surface.

The paper is organised as follows: we begin by introducing the data and methods that we use throughout, and then evaluate the neXtSIM model's general performance for a free run from November 2018 to June 2020 in terms of concentration/extent,

thickness and drift. This free run uses hindcast forcing fields. We then evaluate the neXtSIM-F forecast platform for the same period, when we assimilate concentration but use forecast forcing fields.

## 2 Data sources

### 2.1 Forecast ocean forcing from TOPAZ4

The official European forecast for the Arctic is developed and run by the CMEMS Arc-MFC (Arctic Monitoring and Forecasting Centre)[5]. This uses the TOPAZ system (Simonsen et al., 2018; Sakov et al., 2012), which uses version 2.2.37 of the Hybrid Coordinate Ocean Model (HYCOM) (Bleck, 2002). In the current version (4) of TOPAZ, HYCOM is coupled to a sea ice model derived from version 4.1 of the Community Ice CodE (CICE: Hunke and Lipscomb, 2010); ice thermodynamics are described in Drange and Simonsen (1996), while the dynamics are based on the visco-plastic (VP) sea ice rheology (imple-

mented with the elastic-viscous-plastic (EVP) solver of Hunke and Dukowicz, 1997). The model's native grid covers the Arctic and North Atlantic Oceans and has a horizontal resolution of between 11 and 16 km. TOPAZ4 uses the Ensemble Kalman filter method (EnKF; Sakov and Oke, 2008) to assimilate remotely sensed sea level anomalies, sea surface temperature, sea ice concentration, sea ice thickness and Lagrangian sea ice velocities (the latter two in winter only), as well as temperature and salinity profiles from Argo floats and ice-tethered profilers. Data assimilation is performed weekly.

To force neXtSIM, we use the following daily-averaged variables from TOPAZ: the sea surface $(0-3\,\text{m})$ ocean velocity, temperature and salinity (SST and SSS, respectively), and the mixed layer depth (MLD). We give more details of how they are used in Section 3.1 below.

### 2.2 Forecast atmospheric forcing from ECMWF

For our forecast demonstration, we use the latest version (Cycle 45r1) of the Integrated Forecast System from ECMWF (IFS:

Owens and Hewson, 2018) to provide atmospheric forcing fields to neXtSIM. It consists of an atmospheric model coupled to the NEMO 3.4 ocean model (Nucleus for European Modelling of the Ocean), the LIM2 (Louvain-la-neuve Sea Ice Model) sea ice model, the ECWAM (ECMWF WAve Model) wave model, and a land surface model (HTESSEL). Its spatial resolution is $0.1°$ and while its temporal resolution is 1 h we update our forcing fields less frequently (every 6 h).

The variables we use are the 10-m wind velocity, the 2-m air and dew point temperatures (the latter is used to determine the

specific humidity of air for the latent heat flux calculation), the mean sea level pressure, the long- and short-wave downwelling radiation, and the total precipitation (this becomes snow if the 2-m air temperature is below $0°$C).

---

[5]Three institutes contribute to the Arc-MFC: the Nansen Environmental and Remote Sensing Center, the Norwegian Meteorological Institute and the Norwegian Institute for Marine Research.

## 2.3 Sea ice concentration products from OSI-SAF

OSI-SAF provides estimates of sea ice concentration derived from the Special Sensor Microwave Imager Sounder (SSMIS) radiometer (Tonboe et al., 2016; Tonboe and Lavelle, 2016; Lavelle et al., 2017) and from the Advanced Microwave Scanning Radiometer 2 (AMSR2: Lavelle et al., 2016a, b; Tonboe and Lavelle, 2015). The SSMIS algorithm uses the 19 GHz frequency (vertically polarized, footprint size about 56 km) and the 37 GHz frequency (both vertically and horizontally polarized, footprint size about 33 km). The AMSR2 algorithm uses three frequencies: 18.7, 36.5 and 89 GHz (also in vertical and horizontal polarizations) with footprints from 22 to 5 km). The AMSR2 data are presented on a 10-km grid, and we chose this product over the higher resolution (3.25 km) ASI product as we found it less noisy near the ice edge. These products are available daily within 12 hours after acquisition and processing so it is possible to assimilate this data in operational forecasts. However, the file for the day before the bulletin date (the day the model is run) doesn't arrive early enough to be assimilated in our daily run, which is launched at 03:00 (Central European Time). Therefore, we use the file from two days before the bulletin date.

As specified in the validation reports cited above the SSMIS has lower resolution ice concentration but has the advantage of higher accuracy, while the AMSR2 algorithm has higher resolution but also higher uncertainties. In order to combine the advantages of these products we generated a blended product that was used for assimilation during the forecasts. Blending was performed with a weighted average of the two products (using the errors in the products to calculate the weights):

$$c_{\mathrm{osisaf}} = \frac{c_{\mathrm{ssmis}}\sigma_{\mathrm{ssmis}}^{-2} + c_{\mathrm{amsr2}}\sigma_{\mathrm{amsr2}}^{-2}}{\sigma_{\mathrm{ssmis}}^{-2} + \sigma_{\mathrm{amsr2}}^{-2}}, \tag{1}$$

where $c$ denotes sea ice concentration and $\sigma$ denotes the concentration uncertainty.

Sea ice extent, used as an evaluation metric of the model, was calculated from the concentration product as a sum of areas of all pixels within the model domain with concentration above 15%. Sea ice extent uncertainty was calculated as a difference between the extents calculated from the sum of concentration and uncertainty and concentration alone.

We use both SSMI and AMSR2 products for assimilation by the forecasts, but only SSMI for evaluation of the free run and our forecasts. This was because we found the AMSR2 product somewhat inconvenient due to missing sections of data, which made our evaluation statistics quite noisy. (OSI-SAF SSMI is therefore not an independent validation dataset for the forecasts.)

## 2.4 Sea ice drift from OSI-SAF

We use this product for evaluation of both our free run and our forecasts. It is not assimilated. To produce it, low-resolution ice drift datasets are computed on a daily basis from aggregated maps of passive microwave (e.g. SSM/I, AMSR-E) or scatterometer (e.g. ASCAT) signals (all channels are used) using the continuous maximum cross-correlation method (CMCC: Lavergne et al., 2010; Lavergne and Eastwood, 2010; Lavergne, 2010). Daily 48-hour ice drift vectors can be obtained at a spatial resolution of 62.5 km. As part of our evaluation we apply a filter on the uncertainty given in the product to remove less accurate observations. We usually take the maximum allowed 2-day drift uncertainty to be 20 km, which allows a reasonable sample size of vectors to compare the model to, and right throughout the year. However, it is useful to sometimes focus on the more precise observations when considering winter drift. Therefore in those cases we apply a stricter filter, where the 2-day drift

uncertainty is less than 2.5 km. This completely excludes the summer period of May to September (RMS uncertainty is about 12 km), since surface melting and a denser atmosphere preclude the retrieval of precise information. From October to April, we can still retain about 75% of the observation vectors after using this threshold, with the removed vectors generally being close to the ice edge, coast or the north pole. The error is higher in these regions as the sub-images on which the CMCC method is applied must be reduced to limit them to being inside regions where there actually is ice (Lavergne and Eastwood, 2010). In the case of the north pole, there are fewer observations there, while the vectors in the MIZ have especially high uncertainties (sometimes up to 12 km) due to the high velocities in those regions, combined with the relatively long time interval over which the drift is calculated.

In order to compare neXtSIM drift to this product more accurately, every day at 12:00 we seed synthetic Lagrangian drifters at the grid points of the OSI-SAF drift product and advect them for 48 hours according to the ice velocity in the model. That is, their drift from the original position is updated every model time-step ($\Delta t$)

$$\mathbf{d}_{\mathrm{mod}}(\mathbf{x}, t^n + 1) = \mathbf{d}_{\mathrm{mod}}(\mathbf{x}, t^n) + \mathbf{u}(\mathbf{x}, t^n)\Delta t,$$

where $\mathbf{u}$ is the ice velocity in the model. (Note that the drift $\mathbf{d}_{\mathrm{mod}}$ is a global field.) At output time, and when the model mesh needs regridding due to it becoming too deformed, the drifter positions are updated with

$$\mathbf{x}_{\mathrm{mod},i}(t^{n+1}) = \mathbf{x}_{\mathrm{mod},i}(t^n) + \mathbf{d}_{\mathrm{mod}}\big(\mathbf{x}_{\mathrm{mod},i}(t^n), t^{n+1}\big),$$

and $\mathbf{d}_{\mathrm{mod}}$ is reset to zero. This step, which requires $\mathbf{d}_{\mathrm{mod}}$ to be interpolated, is done as little as possible for the sake of model performance and to avoid the accumulation of interpolation error. The total drift after 48h is then compared to the OSI-SAF drift product.

## 2.5   Sea ice thickness from CS2-SMOS

We use the CS2-SMOS sea ice thickness product (version 2.3: Ricker et al., 2017) to initialise our free run and forecast (this is done once only, and is not to be confused with our daily assimilation which corrects the initial conditions of individual forecast bulletins) and for long-term evaluation of our free run. This product is a daily hybrid product which combines thickness estimates from the CryoSat-2 (CS2) altimeter (more reliable for thicker ice ($\gtrsim 0.5$ m) and from SMOS (Soil Moisture and Ocean Salinity: better for thin ice). The CS2 altimeter tracks are somewhat sparse so optimal interpolation (OI) is used to fill the gaps between them and the areas of thin ice. For this reason each file covers a 7-day period. The OI method requires a background field to be created with full coverage, and that is independent of the target week so it is created from the CS2 values for the two weeks ahead and behind the target week, and from the SMOS values for the day before the target week.

The errors from this approach can be particularly high in coastal areas of thick ice (for example north of Greenland and Canada), which are too thick to be measured by SMOS, but may be only covered by altimeter tracks every 2–3 weeks. For these gaps in coverage, the product uses the background field.

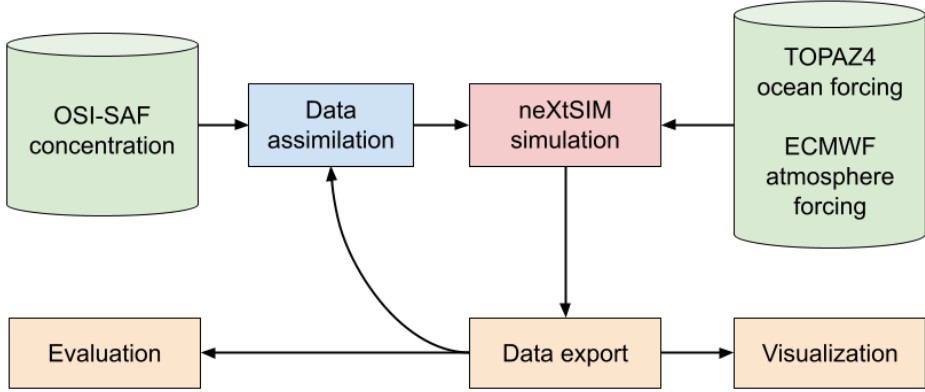

**Figure 1.** Overall scheme of the neXtSIM-F forecasting platform. Blue blocks: pre-processing steps; red block: running of the model core; yellow blocks: post-processing steps; green blocks: input data. Note that we only assimilate OSI-SAF SSMI and AMSR2 concentrations.

## 3 Description of the forecast platform

Figure 1 gives an overview of the neXtSIM-F platform, showing the steps that are run daily automatically. First, initial conditions for the forecast are set by taking the restart file created by the previous day's run and updating them according to the most recent OSI-SAF concentrations (SSMI and AMSR2). The model is then run for 8 days using forecast ocean and atmospheric

forcings — the first day of simulation is an analysis, while the last seven days are forecasts. Finally, some post-processing steps are run: exporting of data (e.g. preparing the CMEMS files and uploading them to the CMEMS Diffusion Unit), creating some visualisations and doing evaluations (where possible). Not shown in the figure are steps that were done only once: initial generation of the model mesh, and initialization of the model fields from the CS2-SMOS thickness product. These steps are explained in more detail in the sections below.

**3.1 The neXtSIM model**

neXtSIM is a stand-alone sea-ice model which can use winds and currents from a variety of atmospheric and oceanic models (hindcasts or forecasts). This makes it quite flexible and light to run and therefore ideal for a forecasting context. Its dynamical core is the new Brittle-Bingham-Maxwell (BBM) rheology (Ólason et al., in prep.). (The version of the BBM rheology corresponding to the results in this paper is also summarised in Appendix A.) Rampal et al. (2016) presented results using a

previous version of neXtSIM including an elasto-brittle (EB) rheology as described in Bouillon and Rampal (2015), showing good agreement with observed drift and concentration in particular. More recently, Rampal et al. (2019) showed the ability to reproduce the characteristic multi-fractal scalings of deformation when using the previous version of neXtSIM, which included the Maxwell-elasto-brittle (MEB) rheology (Dansereau et al., 2016). The key contribution of the MEB rheology was the addition of a viscous dissipation of stress in areas where the ice is damaged, allowing it to move more freely. However, in

longer term simulations the MEB showed unrealistic pile-up of ice particularly along the north-east coast of Greenland and the

north-west coast of Svalbard. This led to the further addition of a frictional element to the rheology (Ólason et al., in prep.), which provides some resistance to compression (up to a threshold). This framework, of a spring in series with a composite element made up a dashpot and fraction element in parallel, is called BBM.

The dynamical equations are solved with a finite element method on a Lagrangian (moving) triangular mesh. The code is currently a parallelised C++ code, used by Rampal et al. (2019), and presented by Samaké et al. (2017). Momentum input comes from the wind and ocean stresses (a turning angle of $25°$ is applied to the ocean velocity from the ocean forcing), the Coriolis force and sea surface slope, and there is also a basal stress applied at the bottom of the ice when it becomes grounded. For this basal stress we follow the scheme of Lemieux et al. (2016b), using the parameters $k_1 = 10$, $k_2 = 15 \, \mathrm{Pa \, m^{-1}}$, $\alpha_b = 20$, $u_0 = 5 \times 10^{-5} \mathrm{m \, s^{-1}}$.

There is also a thermodynamic component of the code and beneath the ice is a slab ocean with three variables: temperature, salinity and thickness. The temperature and salinity are modified by the heat and salinity fluxes determined by the thermodynamical model as ice melts and freezes and as the model interacts with the atmosphere. They are relaxed towards the SST and SSS from TOPAZ over a time scale of about one month, while the thickness of the slab ocean is taken directly to be the MLD of TOPAZ. This varies spatially and evolves with time according to the forecast from TOPAZ. The thermodynamical model is a three-category model (detailed in Rampal et al., 2019, Appendix A): open water, newly-formed ice (treated as one ice layer and one snow layer; Semtner, 1976) and older ice (treated as two ice layers and a snow layer; Winton, 2000).

The older ice is characterised in the model by concentration, $c$, and thickness averaged over the entire cell (effective thickness or, in other words, volume per unit area), $h$. The absolute thickness of ice can be computed as the ratio: $H = h/c$; there is also an effective snow thickness, $s$. The young ice has concentration $c_y$, effective thickness $h_y$, and snow thickness, $s_y$. The absolute thickness of young ice $H_y = h_y/c_y$ is constrained so that $H_{\min} \leq H_y \leq H_{\max}$. If $H_y < H_{\min}$, $c_y$ is reduced to $c'_y = (H_y c_y)/H_{\min}$; if $H_y > H_{\max}$, some ice is moved to the older ice category. During this process $H_y$ is reduced to $H'_y = H_{\max}$, while $c_y$ is also reduced to $c'_y = c_y(H_{\max} - H_{\min})/(H_y - H_{\min})$. This concentration reduction is intended to give some lateral decrease in young ice volume and not just vertical. The corrected values for $h_y$ and $s_y$ are $h'_y = c'_y H_{\max}$ and $s'_y = c'_y(s_y/c_y)$; the corresponding properties for the older ice $h$, $c$ and $s$ are then increased in a ice-and-snow-volume-conserving manner. The values that we used for the thresholds on the absolute thin ice thickness are $H_{\min} = 0.05 \, \mathrm{m}$ and $H_{\max} = 0.275 \, \mathrm{m}$.

The domain we present simulations for is a pan-Arctic one, with a resolution of about 7.5 km. A 7-day forecast for this domain runs in about 30 minutes using 16 processors.

### 3.2 Initialization of the model fields

Before the model can be run it has to be initialized. (Note this is only done once and is not to be confused with the daily assimilation, which modifies the initial conditions for each new forecast). First, the triangular mesh for the destination domain is generated with Unref (a component of the open-source mesh-generation library GMSH: Geuzaine and Remacle, 2009) and using the Global Self-consistent, Hierarchical, High-resolution Shoreline Database (GSHHS) (Wessel and Smith, 1996). Then model variables like the ice concentration, the ice and snow thicknesses, and the temperature and salinity of the slab ocean

are initialized. We use the CS2SMOS product (sec. 2.5) to set the ice concentration and thickness and we use the simulated sea surface temperature and salinity from TOPAZ4 (sec. 2.1). The ice concentration in the CS2SMOS product comes from the low-resolution OSI-SAF product (SSMIS) which is known to be too low in areas with low thickness (Ivanova et al., 2015). Therefore total sea ice concentration ($c_{\mathrm{tot}}$) is calculated by increasing the observed SIC according to the empirical formula:

$$5 \quad c_{\mathrm{tot}} = \frac{1}{a_0}\mathrm{csch}\left(\frac{h_{\mathrm{cs2smos}} - a_1 a_2}{2a_1}\right) \times c_{\mathrm{cs2smos}}, \tag{2}$$

where $c_{\mathrm{cs2smos}}$ and $h_{\mathrm{cs2smos}}$ are respectively the concentration and effective thickness in the CS2SMOS product, and $a_0 = 0.9569$, $a_1 = 0.06787$ and $a_2 = 0.4255$ are parameters fitted from the observations (personal communication with Thomas Lavergne).

As mentioned above the model has two ice categories - young ice and older ice with different rheological behaviour. At the initialization step young ice $c_y$ is set to comprise 20% and the older ice $c$ 80% of the total SIC. If the observed absolute thickness $h_{cs2smos}/c_{\mathrm{tot}}$ is below the young ice upper thickness limit $H_{\max}$, then thickness is distributed between young and older ice in the same proportion, otherwise young ice thickness is calculated as $h_y = c_y H_{\max}$ and for the older ice: $h = h_{\mathrm{cs2smos}} - h_y$. It was identified that the model has little sensitivity to the fraction of young ice and it may vary within 5 - 30%.

The temperature and salinity of the slab ocean is taken to be equal to the TOPAZ4 surface forecast, while the ice velocity and damage are set to zero.

## 3.3 Assimilation of concentration

The assimilation is performed before each forecast run using the data insertion method — an updated (analysis) concentration ($c_{\mathrm{tot}}^{\mathrm{a}}$) is calculated as a function of the forecast variable ($c_{\mathrm{tot}}^{\mathrm{f}}$) and observations ($c_{\mathrm{osisaf}}$), the blended SSMIS/AMSR2 concentration from equation (1). Other variables (particularly ice and snow thicknesses and the SST of the slab ocean) are adjusted for consistency, and the simulation is then restarted using the updated variables and the model is run for 8 days to provide one day of hindcast and a 7-day forecast. The assimilation is performed on the model mesh — the satellite observations, originally provided on a regular grid, are linearly interpolated onto the centers of the mesh elements so that they can be directly compared with the neXtSIM prognostic variables.

The concentration update is done as follows. A target concentration is calculated using a weighted average approach:

$$25 \quad c_{\mathrm{target}} = \frac{c_{\mathrm{tot}}^{\mathrm{f}}\sigma_{\mathrm{model}}^{-2} + c_{\mathrm{osisaf}}\sigma_{\mathrm{osisaf}}^{-2}}{\sigma_{\mathrm{model}}^{-2} + \sigma_{\mathrm{osisaf}}^{-2}}. \tag{3}$$

The uncertainties of the observed concentration ($\sigma_{\mathrm{osisaf}}$) are obtained from the input products (the root mean square of the SSMIS and AMSR2 errors), while the value for the uncertainty in the forecast concentration is set to 0.3 and can also be thought of as introducing a time scale (in days) of $1 + \sigma_{\mathrm{osisaf}}^2/\sigma_{\mathrm{model}}^2$ for relaxation towards the observations that depends on the relative uncertainties. Assimilation of concentration is performed in all elements that have valid observations.

In the pack, the total (young and old ice) model concentration $c_{\mathrm{tot}}$ is generally close to 100% (except inside leads and cracks), while the OSI-SAF concentration can be around $90 - 95\%$. We found that using $c_{\mathrm{target}}$ as the actual update $c_{\mathrm{tot}}^{\mathrm{a}}$ produced too much of a drop in the pack concentration, lowering the internal stress to near zero and allowing too much drift. This quickly led

to rapid build up of very thick ice in unusual places. Therefore we had to take a quite conservative approach to our correction and only change the model to make sure the ice mask (the part of the domain where the concentration is higher than 15%) was correct (this is a kind of assimilation of extent):

$$c_{\text{tot}}^a = \begin{cases} 0 & c_{\text{target}} < 0.15, \\ c_{\text{target}} & c_{\text{tot}}^{\text{f}} < 0.15 \text{ and } c_{\text{target}} \geq 0.15, \\ c_{\text{tot}}^{\text{f}} & \text{otherwise.} \end{cases} \tag{4}$$

After calculating the updated variables as specified in (4) the fractions and thicknesses of young and older ice are calculated in the same way as in the initialisation procedure (see sec. 3.2).

     Once all the assimilation steps have been performed the model fields are checked for consistency with each other and corrected if necessary. First, the concentration of young ice is reduced in the elements where the total concentration exceeds 100%. Second, the volume of ridged ice, damage and the ice and snow thicknesses are set to zero in the added ice. Lastly, we
need to correct the SST of the slab ocean. Where new ice is added during assimilation it is set to the freezing point, but if ice is removed then the situation is more complicated. Then we proceed as in the following section.

### 3.4   Compensating heat flux

One of the side effects of assimilation is that the heat balance in the model is disturbed: reducing the concentration opens more water and depending on the temperatures, atmospheric humidity and ocean salinity provided by the forcing, the heat flux out
of the ocean can increase dramatically causing the ice to freeze up again very fast. This effect is strongest when the atmosphere is very cold. Therefore a compensating heat flux is added to the total ocean heat flux in order to keep the heat balance and prevent refreezing of ice where it was removed by assimilation, thus prolonging the effect of assimilation. If $c_{\text{tot}}^{\text{f}}$ and $c_{\text{tot}}^{\text{a}}$ are respectively the total concentrations before and after assimilation, a compensating heat flux ($Q_{\text{comp}}$) is calculated if ice was removed — i.e. if $c_{\text{tot}}^{\text{a}} < c_{\text{tot}}^{\text{f}}$ — according to the following formula

$$Q_{\text{comp}} = Q_{\text{ocean}}\left((c_{\text{tot}}^{\text{a}}/c_{\text{tot}}^{\text{f}})^n - 1\right) \tag{5}$$

where $Q_{\text{ocean}}$ is the total heat flux from the ocean (sum of flux from the ocean to sea ice and to the atmosphere), and $n$ is a parameter controlling the strength of correction. The function was chosen so that $Q_{\text{comp}}$ is zero when the concentration update is zero, and $Q_{\text{comp}} = -Q_{\text{ocean}}$ when $c_{\text{tot}}^{\text{a}} = 0$, i.e. all the ice was removed by assimilation. With $n = 1$ the reduction of $Q_{\text{comp}}$ from 0 to $-Q_{\text{ocean}}$ is linear and with $n > 1$ it becomes steeper for values of $c_{\text{tot}}^{\text{a}}$ closer to $c_{\text{tot}}^{f}$. We use $n = 4$ for the runs
presented here, as this gives a suitably strong heat flux for a modest reduction in concentration.

### 4   Results

We begin with an evaluation of a free run over the 20-month period from 1 Nov 2018 to 30 June 2020. This shows the general strengths and weaknesses of the model. We then evaluate the performance of the forecast system over this period to show the improvements gained by the assimilation of OSI-SAF concentration.

## 4.1 Evaluation of free model run

In this section we demonstrate that the model is generally able to reproduce the overall drift, concentration, and thickness patterns in observations. For all comparisons we average the model fields in time over an appropriate time window (in practice 1, 2 or 7 days), apply some spatial smoothing (being guided by the size of the satellite footprint), and interpolate onto the observation grid. For scalar variables, we define bias as $\langle V_{\mathrm{mod}} - V_{\mathrm{obs}} \rangle$ (with $\langle \cdot \rangle$ defining the spatial mean over pixels where both model and observation are defined, and where either model or observations have ice.) We also define RMSE as $\langle (V_{\mathrm{mod}} - V_{\mathrm{obs}})^2 \rangle^{1/2}$. For the ice extent, we define bias in extent as $A_{10} - A_{01}$, where $A_{10}$ is the total area of pixels where neXtSIM predicts the presence of ice (total concentration greater than 15 %) but the observation has no ice, while $A_{01}$ is the total area of pixels where neXtSIM predicts no ice but the observation does have ice. Instead of an RMSE, for the ice extent we define IIEE (Integrated Ice Edge Error, Goessling et al., 2016) as $A_{10} + A_{01}$. Thus the IIEE is always positive, and the bias is positive if neXtSIM is overestimating the total extent and negative if it is underestimating it. For the ice velocity we define the bias and RMSE as the mean and RMS values of the difference in speed respectively — i.e. $\langle |\boldsymbol{u}_{\mathrm{mod}}| - |\boldsymbol{u}_{\mathrm{obs}}| \rangle$ and $\langle (|\boldsymbol{u}_{\mathrm{mod}}| - |\boldsymbol{u}_{\mathrm{obs}}|)^2 \rangle^{1/2}$. We also define the vector RMSE (VRMSE), as the RMS of the vector difference, $\langle |\boldsymbol{u}_{\mathrm{mod}} - \boldsymbol{u}_{\mathrm{obs}}|^2 \rangle^{1/2}$. Note that these drift errors can only be calculated where both model and observation have ice.

**Table 1.** Accuracy of the free run. Concentration and extent are evaluated against OSI-SAF SSMI; thickness is evaluated against CS2-SMOS; drift against OSI-SAF drift, where observations with reported error up to 10 km/day are considered. Results are 2-monthly-averaged.

| | Concentration, % | | Extent, $10^6 \mathrm{km}^2$ | | Drift, km/day | | |
| --- | --- | --- | --- | --- | --- | --- | --- |
| | Bias | RMSE | Bias | RMSE | Bias | RMSE | VRMSE |
| Nov-Dec 2018 | 4.91 | 9.97 | 0.09 | 0.24 | 0.00 | 3.63 | 5.16 |
| Jan-Feb 2019 | 2.73 | 6.94 | 0.09 | 0.17 | -0.31 | 2.96 | 4.00 |
| Mar-Apr 2019 | 1.41 | 5.38 | -0.03 | 0.18 | -0.40 | 2.77 | 3.65 |
| May-Jun 2019 | -1.65 | 17.84 | -0.15 | 0.81 | -0.91 | 3.19 | 4.19 |
| Jul-Aug 2019 | -2.47 | 18.79 | -0.06 | 1.05 | 0.23 | 3.25 | 4.75 |
| Sep-Oct 2019 | 1.90 | 14.43 | -0.06 | 0.47 | 0.81 | 2.80 | 4.50 |
| Nov-Dec 2019 | 3.47 | 10.86 | -0.11 | 0.41 | 0.32 | 2.96 | 4.53 |
| Jan-Feb 2020 | 2.76 | 6.06 | 0.05 | 0.17 | 0.09 | 3.15 | 4.48 |
| Mar-Apr 2020 | 1.36 | 4.90 | -0.03 | 0.14 | -0.15 | 3.06 | 4.34 |
| May-Jun 2020 | -0.99 | 16.14 | -0.08 | 0.71 | -0.74 | 3.30 | 4.41 |

Table 1 shows 2-monthly summary statistics for observations that are available year round (although they are much less reliable between May and September). Errors which are particularly high are the RMSE in ice concentration and the IIEE for extent between May and August — we will discuss reasons behind this below. The bias in these quantities is quite low, being highest between November and December, reflecting an ice advance which is too fast. The drift on the other hand compares quite well throughout the period, with the VRMSE being generally less than 4.5 km/day and the RMSE in speed less than 3.3 km/day (with the exception of the first two months). The bias is quite low, staying between ±0.8 km/day. The first two

months have a bias of zero, despite their high VRMSE and RMSE, with a too-slow Beaufort Gyre cancelling a too-fast drift in the triangle between the north pole, Svalbard and Franz Josef land.

**Table 2.** Accuracy of the free run for the 2018 – 2019 and 2019 – 2020 winters. Thickness is evaluated against CS2-SMOS; drift against OSI-SAF drift, where only observations with reported error less than 1.25 km/day are considered. Results are 2-monthly-averaged.

| | Thickness, m | | Drift, km/day | | |
|---|---|---|---|---|---|
| | Bias | RMSE | Bias | RMSE | VRMSE |
| Nov-Dec 2018 | -0.09 | 0.26 | -0.13 | 2.62 | 4.06 |
| Jan-Feb 2019 | 0.09 | 0.36 | -0.26 | 2.22 | 3.27 |
| Mar-Apr 2019 | 0.07 | 0.51 | -0.24 | 2.21 | 3.03 |
| Nov-Dec 2019 | 0.04 | 0.40 | 0.19 | 2.59 | 4.11 |
| Jan-Feb 2020 | 0.23 | 0.46 | 0.27 | 2.41 | 3.69 |
| Mar-Apr 2020 | 0.19 | 0.56 | 0.01 | 2.39 | 3.64 |

Table 2 shows the effect of restricting the drift evaluation to regions where the observational error is less than 1.25 km/day. As discussed in Section 2.4 this removes the entire period from May to September, as well as the MIZ, the area around the north pole and areas close to the coast. Without the less accurate observations, the drop in VRMSE is between 0.3 – 1 km/day, with their values now between 3 and 4.1 km/day. However, the less accurate regions are also regions where the model has difficulties with other variables like thickness and extent, so this could also contribute to the reduction in error. Notwithstanding this, the Nov – Dec 2018 period is now much less of an outlier, now being quite close to Nov – Dec 2019. The RMSE in speed is now less than 2.6 km/day while the bias is between $\pm 0.3$ km/day. There is a clear deterioration going from the first to the second winter.

Table 2 also summarises the comparison of model thickness with the CS2-SMOS product. RMSE is lower (0.26 m) in Nov-Dec 2018 since thickness was initialised using CS2-SMOS, and grows to 0.51 m over the first winter. The RMSE grows from 0.4 m to 0.56 m over the second winter. The bias is quite low in the first winter, but about 0.2 m from Jan – Apr 2020.

Figures 2 and 3 compare the mean concentration and extent from neXtSIM and the OSI-SAF concentration product (see 2.3). Since we do not have enough information to create an error model for the concentration observations (for example it cannot be Gaussian due to its restriction to being between 0 and 100%) we could not generate confidence intervals for the mean concentration, but instead plot a shaded area corresponding to $\pm \langle \sigma_{\text{OSISAF}} \rangle^{1/2}$ for reference when considering the bias and RMSE in concentration. The seasonal cycle for mean concentration and extent is captured reasonably well, but the bias is clearly positive in winter and clearly negative in summer. As noted from Table 1 the RMSE in concentration and the IIEE are very high between mid-April and August. The RMSE in concentration is generally comparable to the RMS error level outside these periods.

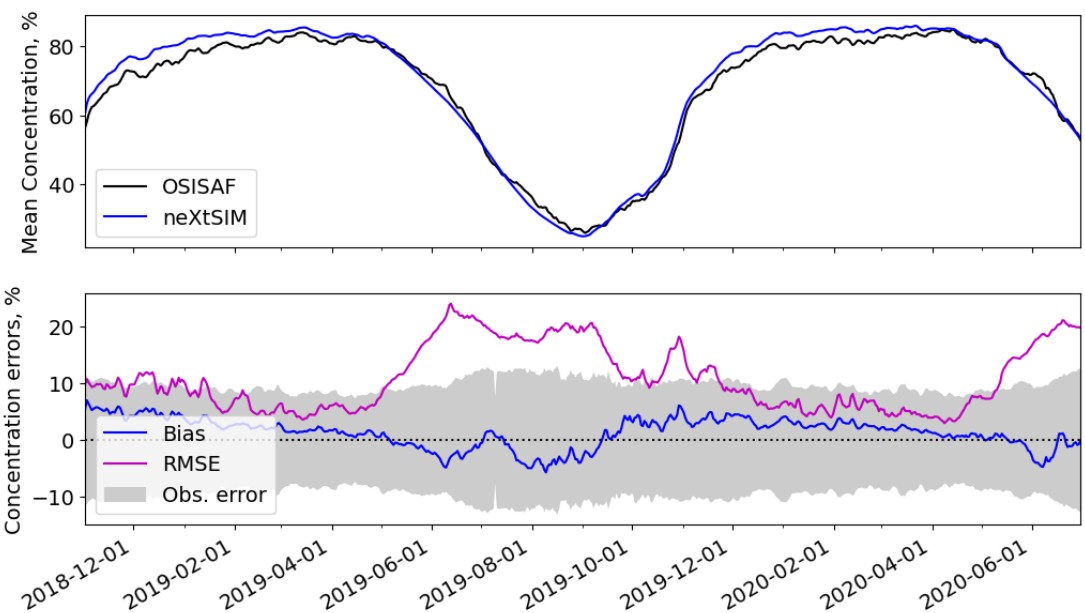

**Figure 2.** Temporal comparison of model and OSI-SAF SSMI concentrations. The shaded area in the lower plot shows the RMS uncertainty of the concentration for reference when considering the errors. The mean concentration is the mean over the ocean points in the domain, while the bias is defined as the mean error (model – observation) over the region where either the model or observations have ice.

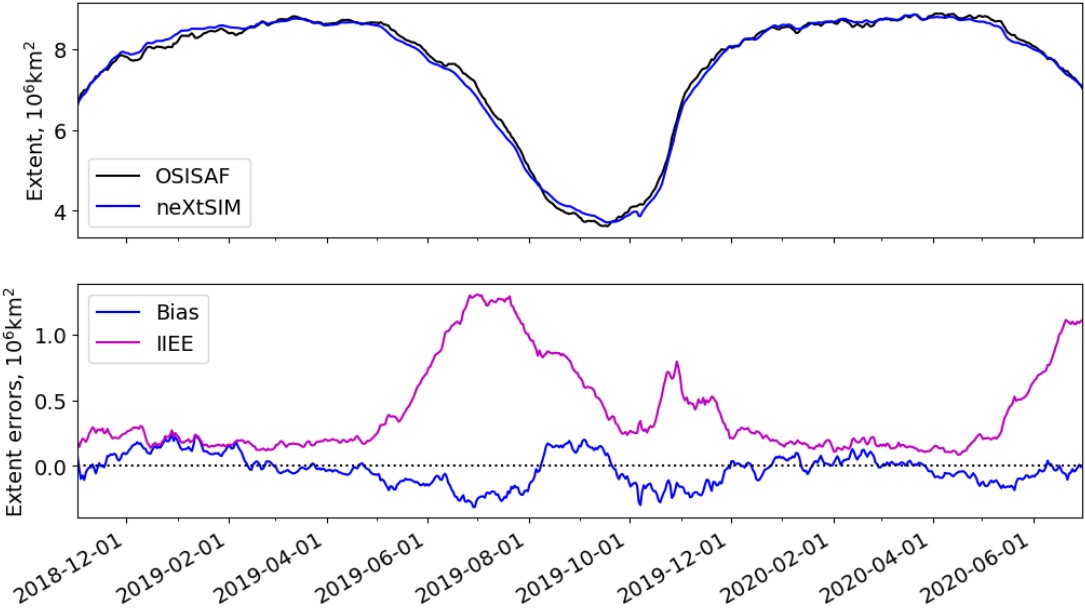

**Figure 3.** Temporal comparison of extents (region where the concentration exceeds 15%) from model and OSI-SAF SSMI concentrations. The bias is defined as modelled extent – observed extent.

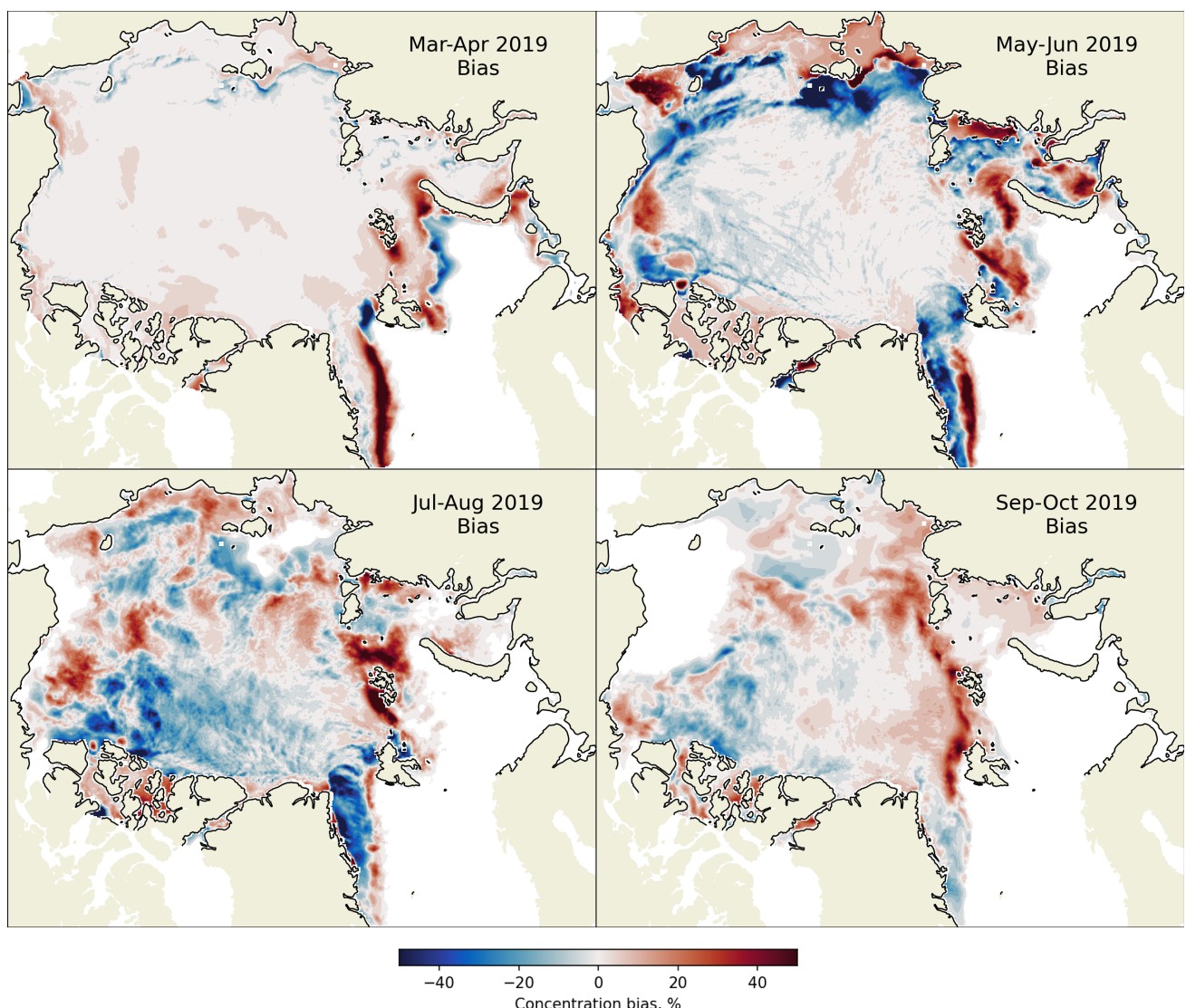

**Figure 4.** Selected maps of the 2-monthly-averaged concentration biases between the free model run and from OSI-SAF SSMI. The bias is defined as model – observation.

In Jan – Feb there is a general overestimation in the pack as the model is approximately 100% with some leads, while the SSMI field has large regions of lower concentration that contribute to the errors (85 – 90%). There is also a general overestimation in ice extent, which is also the case in Nov – Dec, primarily located in the Greenland Sea, and the Barents Seas. The areas north of Novaya Zemlya (around the Santa Anna Trough), around Franz Josef land, and northwest of Svalbard stood out as places where the ice edge was not being located correctly. This continues into Mar – Apr, which can be seen in Figure 4, that shows selected maps of the concentration bias from Mar – Oct 2019. The region of underestimation to the northwest of Svalbard is related to the development of an arch between there and the northeast tip of Greenland. (This is also related to

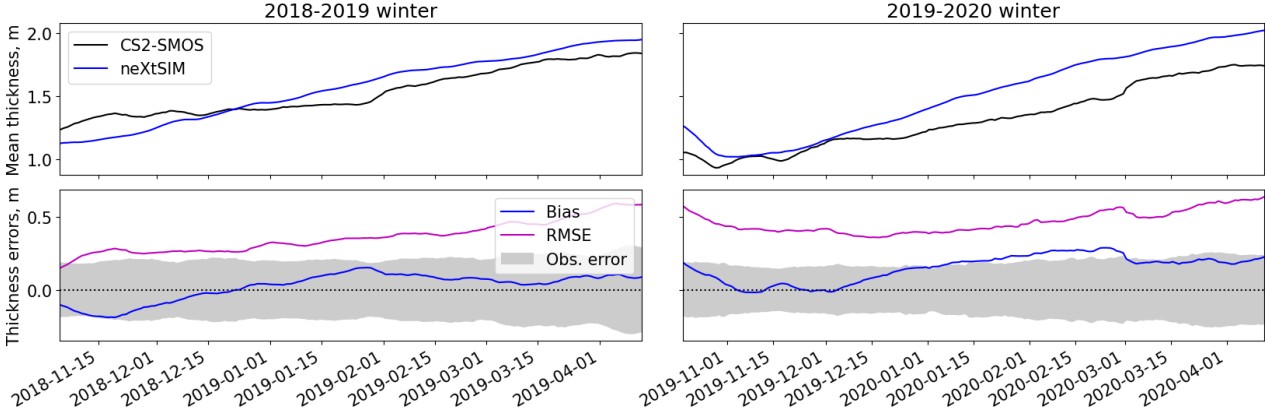

**Figure 5.** Temporal comparison of model and CS2-SMOS thickness for two winters, 2018 – 2019 in the left-hand columns and 2019 – 2020 on the right-hand columns. The shaded regions show the RMS uncertainty in the CS2-SMOS product for reference. The mean thickness is the mean over the ocean points in the domain, while the bias is defined as the mean error (model – observation) over the region where either the model or observations have ice.

relatively large thicknesses in these areas, which increases the capacity of undamaged ice to stay undamaged.) This arching also reduces the ice export through the Fram Strait, the effect of which is seen in the maps for May – Jun and Jul – Aug. In May – Jun we also see that the ice that is in the Greenland Sea is displaced too far to the east, producing a double penalty effect in the RMSE score. This is also a result of the arching in this area — the ice at the corner of Greenland is too thick to be

exported, and so the ice that is exported has detached from the arch away from the coast and has then travelled roughly parallel to the coast without being pushed back towards it. In Jul – Aug the thicker ice has mostly melted and there is less of a dipole situation and more of a clear-cut underestimation.

Another problem area is around the Novo Sibirski Islands and the Laptev Sea. April sees the model opening slightly too far to the north; in May and June there is a strong underestimation to the north of the islands and too much fast ice. The

underestimation is partly from too fast melting and that the fast ice has not detached and flowed into this area.

There is a similar problem to the west of Wrangel Island in the Chukchi Sea — an opening that is too large compared to the observations and too much land-fast ice close to the coast which should be flowing into this region. There is also a large region of overestimation to the east of Wrangel Island, which is actually an artefact of the boundary conditions that we are using at the open Bering Strait boundary. If there is inflow at any open boundaries the value of any tracers is taken to be its value in the

nearest mesh element, and in this case we are getting too much ice being imported through the Bering Strait, which leads to a build-up at this location.

The last things to mention about the Jul – Aug map are the strong overestimation around Franz Josef land, and the dipole situation in the Beaufort sea, which is the result of a too-slow Beaufort Gyre.

However, by Sep – Oct, the situation has improved substantially. The main disagreement is in the ice edge from Severnaya

Zemlya round to the just past Svalbard. This is due to an ice advance that is slightly too fast.

Figure 5 shows time series of mean thicknesses and of thickness errors when compared to CS2-SMOS, which have already been discussed to some extent in the context of Table 2. The mean observed and modelled thicknesses show a steady increase after November, but the modelled value is increasing faster than the observed one. The timing of when the modelled increase starts is close to when the observed increase starts. The lower plots have the RMS uncertainties plotted for reference, and the model bias is generally at a similar level to this uncertainty. The RMSE is about twice this uncertainty in the second winter. We note here that the error levels in the CS2-SMOS product are only the interpolation error, and are thus a lower bound as they don't include uncertainties in the individual CS2 and SMOS products. CS2 in particular is sensitive to the ice and snow densities used or the snow thickness which affect the conversion from freeboard to thickness (Zygmuntowska et al., 2014).

Figure 6 also shows the spatial distribution of the errors. Throughout the winter there is overestimation off the north coast of Greenland round to Axel Heiberg Island. Further to the west there is a thinning (but not an opening) from Axel Heiberg Island to Ellef Ringnes Island which also seems to be related to the westward drift along this coast being too high. This persists throughout the winter but the affected area reduces with time. In Oct – Dec there is a dipole pattern where the ice is too thin in the Beaufort Sea and too thick in the triangle between the north pole, Svalbard and Franz Josef Land. However, in Jan – Apr, this dipole reverses.

The modelled thickness is also quite high around the north of Svalbard, in contrast with the CS2-SMOS product, which has quite thin ice. This build-up contributes to the arching discussed in relation to the concentration errors, reducing the ice export through the Fram Strait. This stems from damaged ice not having enough resistance to compression. In the BBM rheology there is a balance between resisting compression enough to stop the build-up of ice at the coasts and resisting it so much that drift becomes too slow.

Figure 7 shows the drift bias and RMSE of neXtSIM when compared to the OSI-SAF drift product. The shaded area corresponds to $\langle 2\sigma^2_{\text{OSI-SAF}} \rangle^{1/2}$, where the factor of 2 comes from the fact that drift has two components. We also note that if one cell has components $(x_i, y_i)$ with (not-necessarily-Gaussian) noise $(\varepsilon_i, \delta_i)$ added to it (each with zero mean and variance $\sigma^2_i$), the mean-squared drift in the presence of noise,

$$\langle \tilde{d}^2 \rangle = \frac{1}{N}\sum_{i=1}^{N}\langle(x_i+\varepsilon_i)^2+(y_i+\delta_i)^2\rangle = \frac{1}{N}\sum_{i=1}^{N}\langle x_i^2+y_i^2+\varepsilon_i^2+\delta_i^2\rangle = \frac{1}{N}\sum_{i=1}^{N}\left(x_i^2+y_i^2+2\sigma_i^2\right) = \langle d^2\rangle + 2\langle\sigma^2_{\text{OSI-SAF}}\rangle \tag{6}$$

is higher than the noise-free ("true") value $\langle d^2 \rangle$ by $2\langle\sigma^2_{\text{OSI-SAF}}\rangle$. It is tempting to then just subtract this amount from the speed when comparing with the model, but we decided that the uncertainty in the quoted uncertainty could be too influential and also that the model drift has some unknown uncertainty associated with it so we persisted with directly comparing the drift in the OSI-SAF product with the model drift in order to judge when the model is too fast or too slow, and plotted the error level $\langle 2\sigma^2_{\text{OSI-SAF}} \rangle^{1/2}$ for reference. The difference in speed (model speed - observed speed) fluctuates somewhat, but stays largely within the error limits. The RMS error for this product is approximately 1.8 km/day from October to May, but increases to about 4 km/day from May to September. However, the model is starting to show signs of being too slow in April and May 2020. The RMSE in speed ranges between about 3 – 4.5 km/day, which is outside the estimated error for the more accurate colder months. The VRMSE (vector RMSE) is higher than the RMSE by about 1.5 km/day.

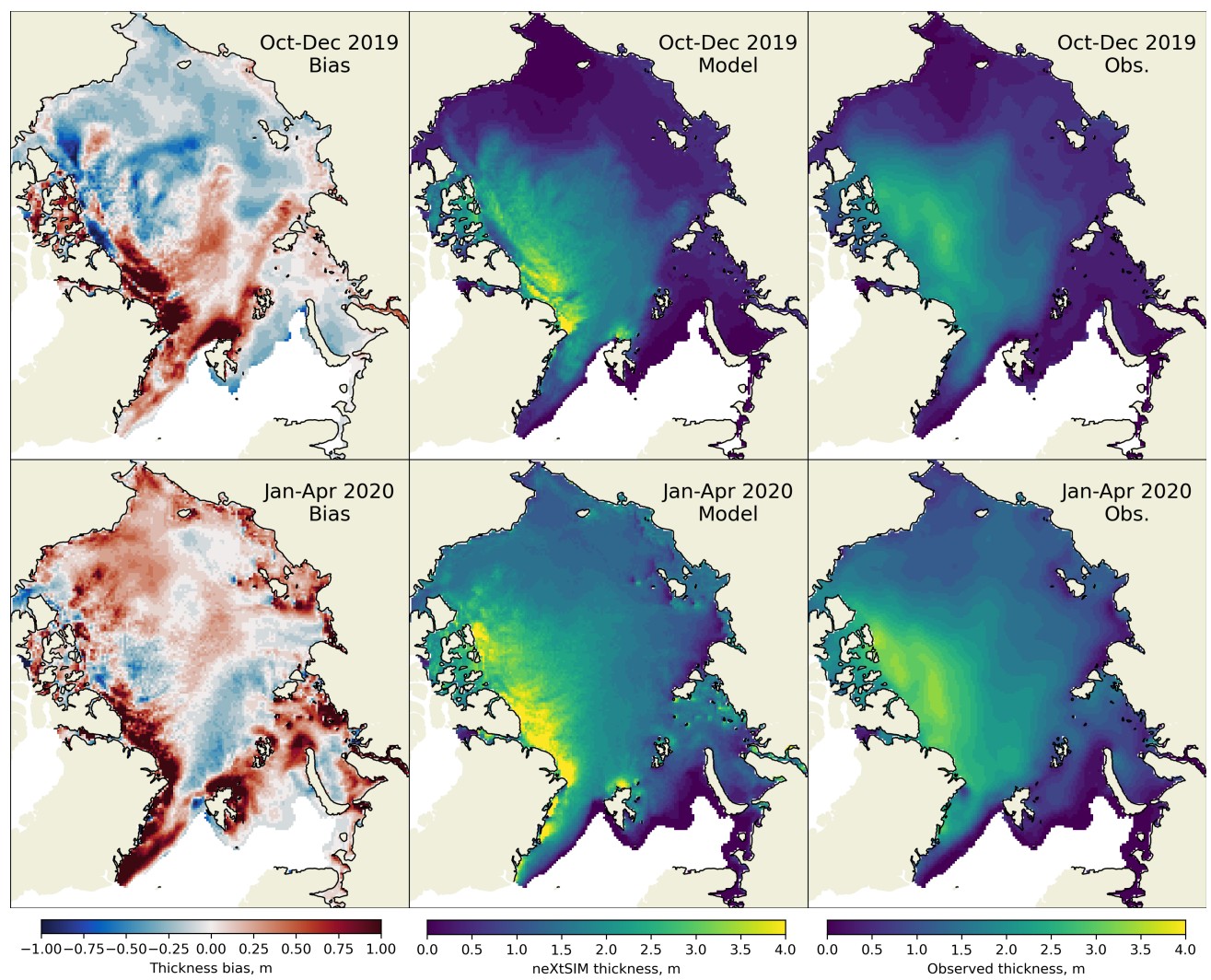

**Figure 6.** Comparisons of time-averaged model and CS2-SMOS thickness for early (top row) and late (bottom row) in the 2019 – 2020 winter. Left hand column: bias maps; centre column: neXtSIM thickness; right hand column: CS2-SMOS thickness. The bias is defined as model – observation.

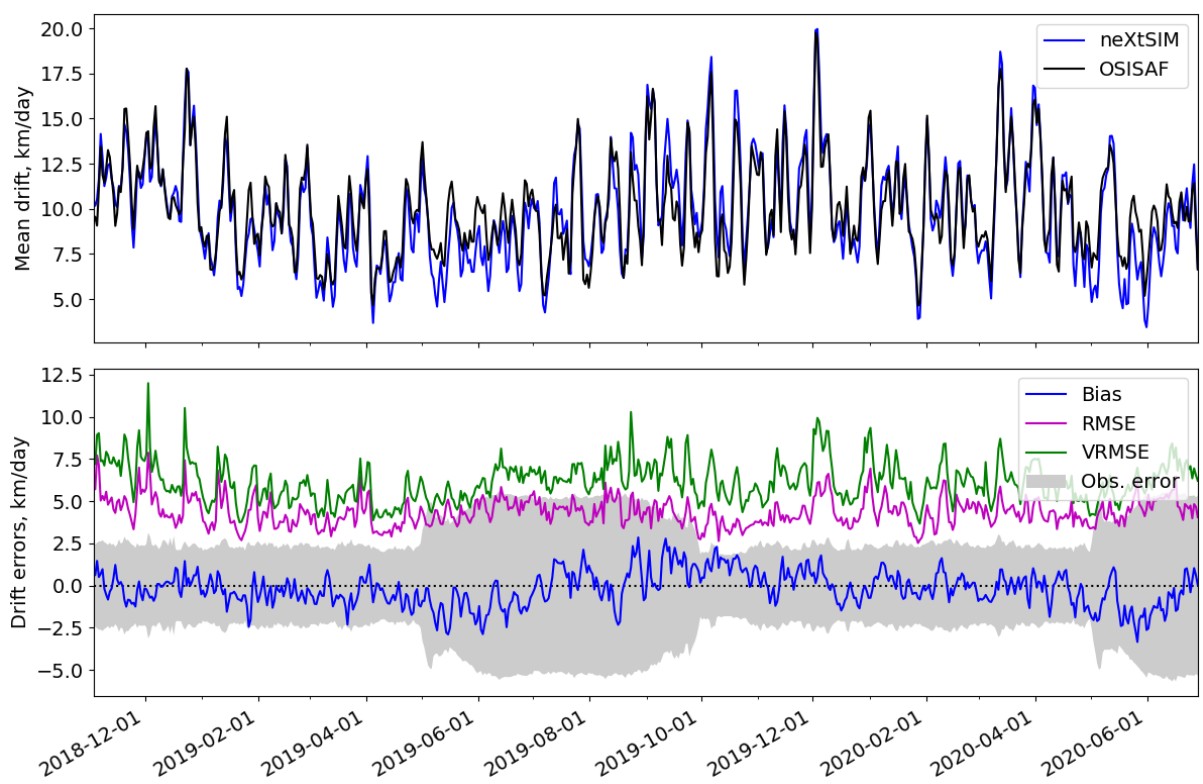

**Figure 7.** Temporal comparison of model and OSI-SAF drift. The shaded region shows the RMS uncertainty in the OSI-SAF drift product. The bias is defined as the mean modelled speed – observed speed. The mean drift and errors are averages over the regions where both model and observations have valid drift vectors. Valid observations here are those that have reported errors less than 10 km/day.

Figure 8 shows the general spatial pattern in the drift errors. The maps for Oct – Dec 2019 show that the general circulation of the model and the observations are agreeing well, with the main feature being the gyre in the central Arctic Ocean. However, this gyre is slightly too fast and there is also too much westward drift along the Canadian archipelago. The observations also show a small gyre in the East Siberian sea which is not apparent in the model drift. The strongest positive bias is in the Kara 5 Sea although the errors in the observations are high there, being both close to the coast and the MIZ. The model is also showing strong positive biases in the Laptev and East Siberian seas where the ice is thinner. On the other hand, the drift in the Beaufort and Chukchi Seas is too slow.

In Jan – Mar 2020, the general pattern in the observations is a strong flow from the Laptev over to the Greenland Sea, with a weaker transpolar drift from the Chukchi Sea joining this outflow. The model also produces the flow from the Laptev Sea, 10 but the flow from the Chukchi Sea is more directed towards the Beaufort Sea than across to the Greenland Sea. This may be contributing to the biases seen in the thicknesses — too thick in the Beaufort Sea while too thin in the region between the Laptev

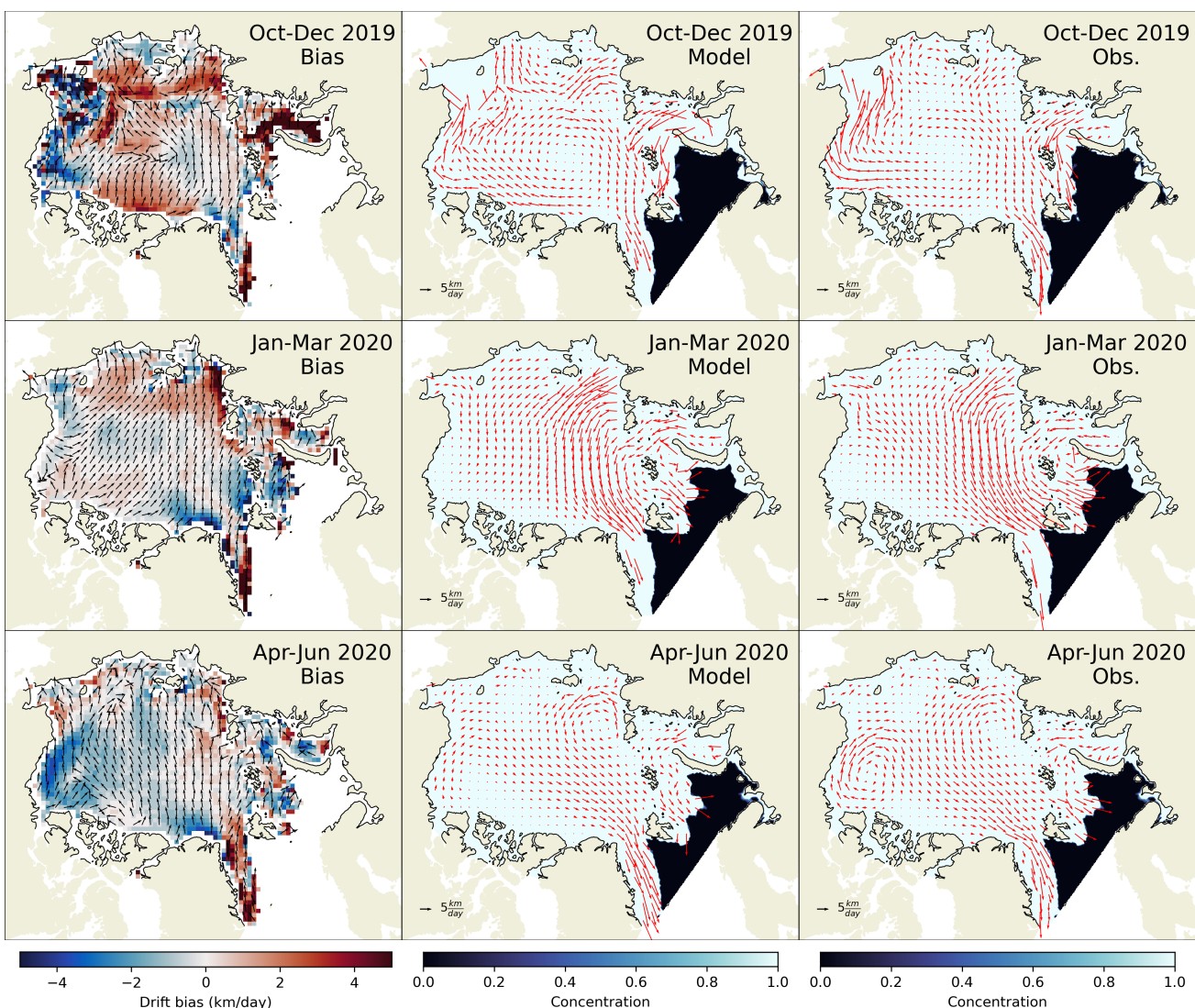

**Figure 8.** Comparisons of 2-monthly averaged model and OSI-SAF drift from the 2019 – 2020 winter. The drift bias colour maps (left column) show the bias in speed, while the directions show the difference between the model and observation directions (arrows pointing up indicate the directions are the same). The central column shows ice velocity vectors from neXtSIM, while the right hand column shows OSI-SAF vectors. The concentration colour maps show the average neXtSIM concentration over the analysis period. The bias is defined as modelled speed – observed speed.

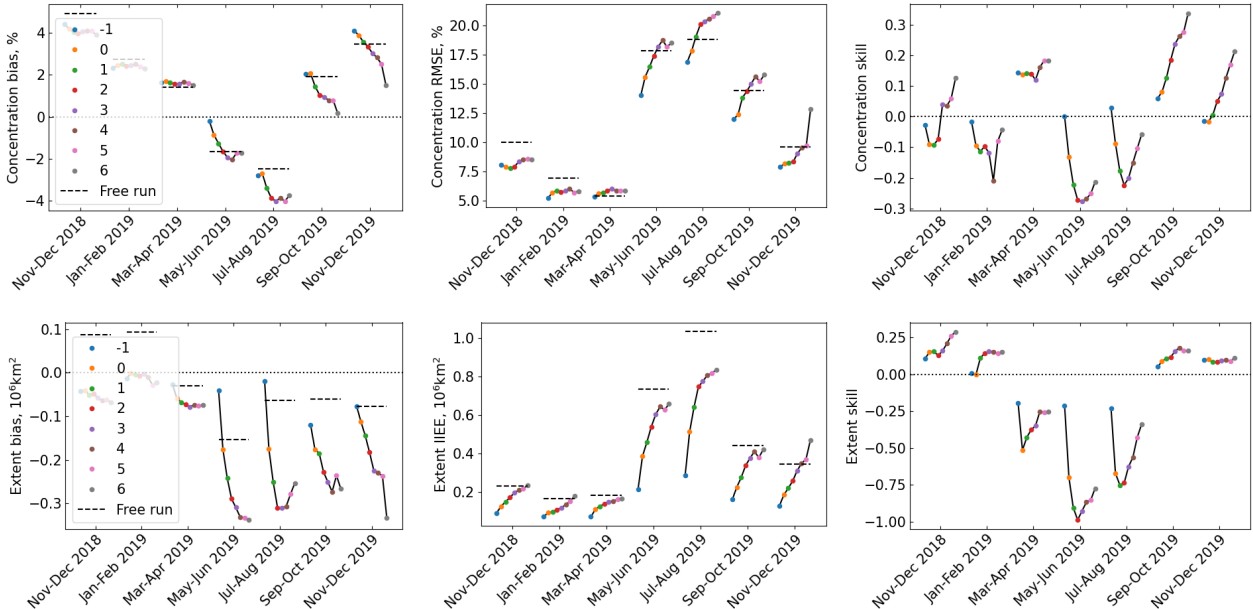

**Figure 9.** 2-monthly-averaged forecast errors grouped by lead time (lead times are indicated in the figure legends). The free run errors for the corresponding periods are plotted as dotted lines for reference on the bias and RMSE/IIEE plots. Upper row: 2-monthly-averaged bias, RMSE and forecast skill for concentration grouped by lead time. Lower row: 2-monthly-averaged bias, IIEE and forecast skill for extent grouped by lead time.

and Greenland Seas. The drift is also starting to become too slow north of Greenland and Svalbard, and this is most likely due to the ice build-up in those areas.

In Apr – Jun 2020 there is a stronger Beaufort Gyre and a similarly strong cyclonic gyre in the Laptev Sea. These connect to form quite a wide stream from the central Arctic to the Greenland Sea. While the model captures the gyre in the Laptev Sea, the Beaufort Gyre is less well-defined in this period, possibly due to the ice being too thick in the Beaufort Sea at this time. As in Jan – Mar, Drift is also too slow to the north of Greenland.

## 4.2 Evaluation of forecasts with assimilation

The performance of the forecasting system with assimilation of concentration was evaluated over the same period as the free run was in Section 4.1 (that is the 20 months from November 2018 to the end of June 2020). In order to reduce computation time, the 7-day forecasts were launched approximately only every seven days with 1-day analyses being launched in between so that assimilation was still performed daily.

The model performance was evaluated against satellite observations of the sea ice SSMI concentration and drift from OSI-SAF (see Sections 2.3 and 2.4). The metrics we used in our assessment were: bias and RMSE in concentration; bias and IIEE in extent; bias, RMSE and VRMSE in drift. A persistence forecast of concentration (the initial concentration, defined for

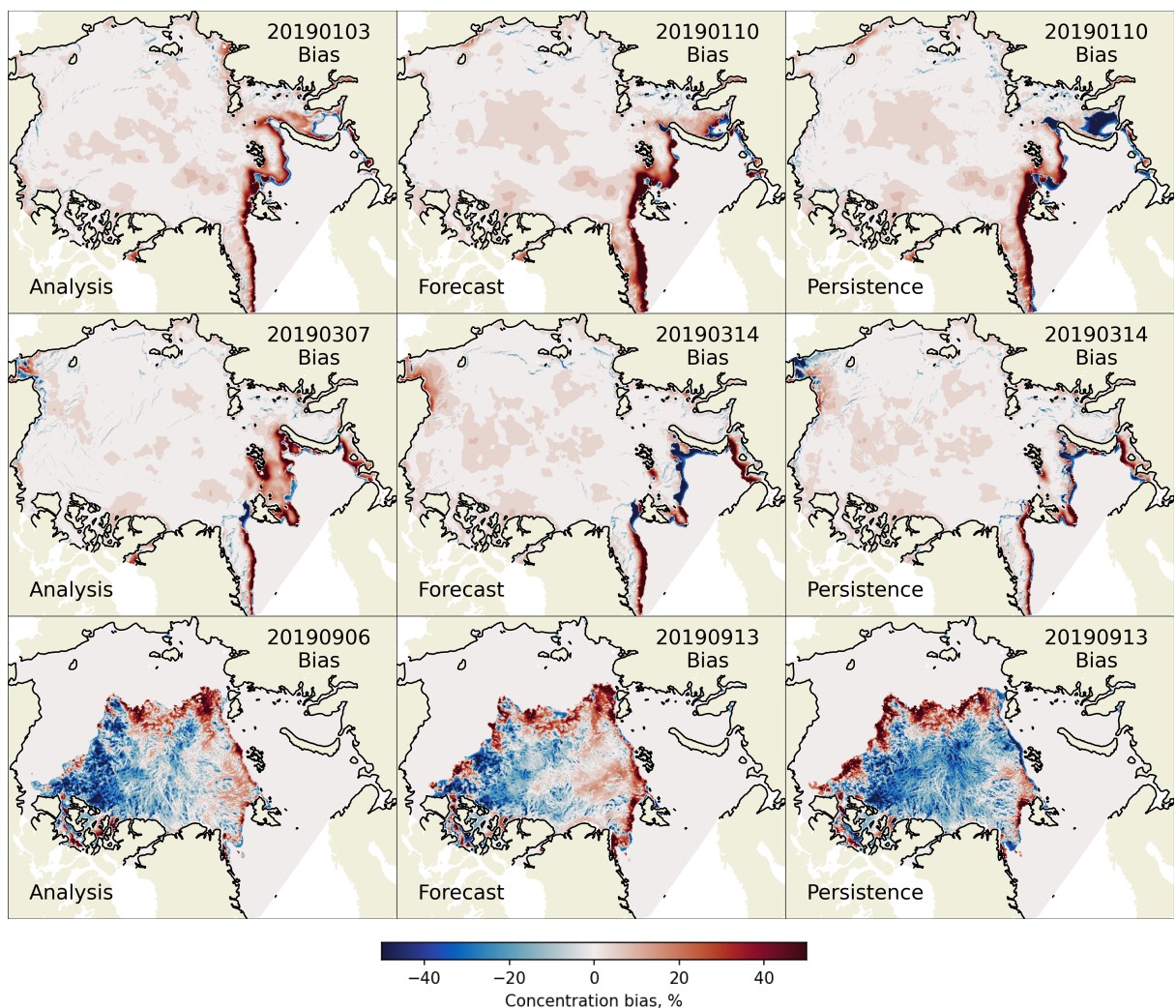

**Figure 10.** Selected maps of daily-averaged biases in concentration compared to OSI-SAF SSMI for three examples of forecasts, starting on 3 January 2019 (upper row, positive skill) and the other on 7 March 2019 (middle row, negative skill) and 6 September 2019 (lower row, positive). The left hand columns show the evaluation for the first day of simulation (lead time -1: analysis or hindcast) while the right hand columns show the evaluation for the eighth day of simulation (lead time 7). The right hand columns show the bias in the persistence forecast for lead time 7. The bias is defined as model – observation.

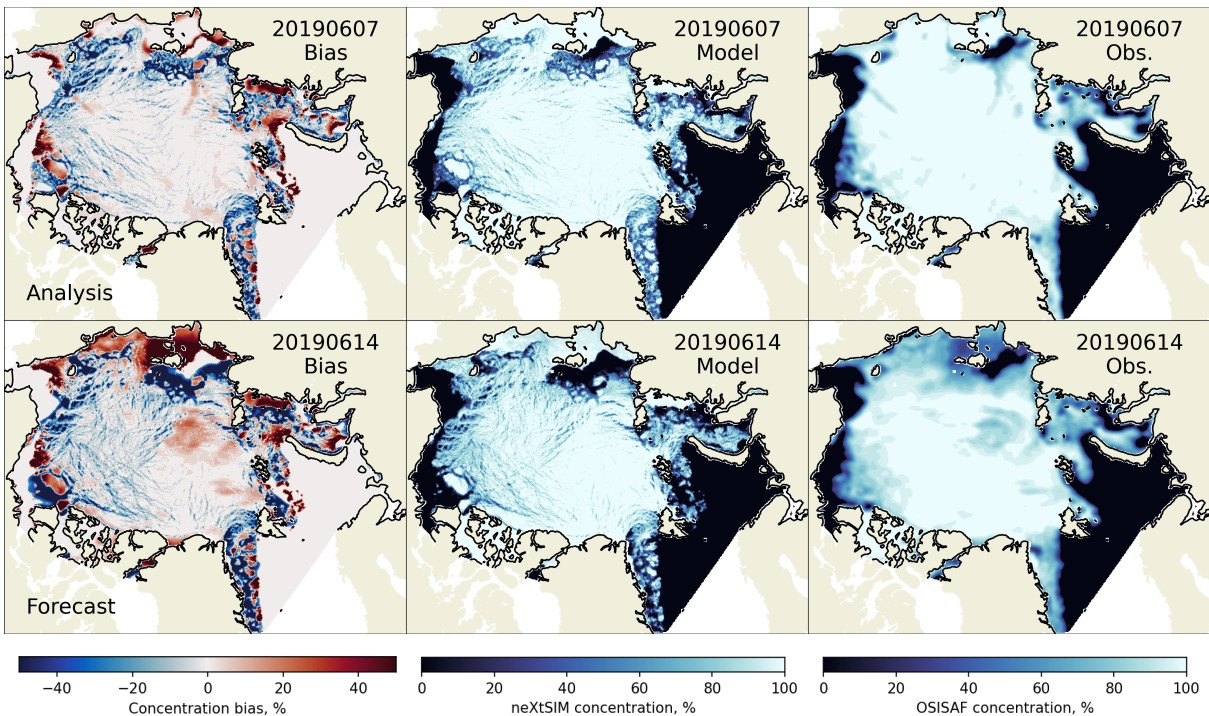

**Figure 11.** Maps of daily-averaged biases (left-hand columns) in concentration compared to OSI-SAF SSMI, and modelled (centre column) and observed (right-hand column) concentrations, for an example forecast starting on 6 June 2019. The upper row shows the comparison for the first day of simulation while the lower row shows the comparison for the eighth day of simulation. The bias is defined as model – observation.

technical convenience as the average of the first hour) was used as a benchmark for concentration and extent, and forecast skill was defined as:

$$\text{Concentration skill} = 1 - \left(\frac{\text{RMSE}}{\text{RMSE}_{\text{persistence}}}\right)^2, \tag{7a}$$

$$\text{Extent skill} = 1 - \frac{\text{IIEE}}{\text{IIEE}_{\text{persistence}}}. \tag{7b}$$

5 Thus, a model that is agreeing perfectly with the observations has skill equal to 1, while it is negative (with no lower bound) if the persistence error is lower than the forecast error. We decided that given the strong dependence of drift on wind, its rapid variability in time would render a persistence forecast for drift quite easy to beat, so for this variable we just present the forecast errors. However, we can make some rough comparisons to the drift errors of other products, even though they use different observations for their evaluation so we can't make a direct comparison. We note that the drift from the TOPAZ

10 forecast generally has a bias in speed of about 2 km/day and a VRMSE of about 5 – 8 km/day Melsom et al. (2018), while Metzger et al. (2017) report an RMS drift speed error of about 5–8 km/day in the Arctic for the GOFS 3.1 system.

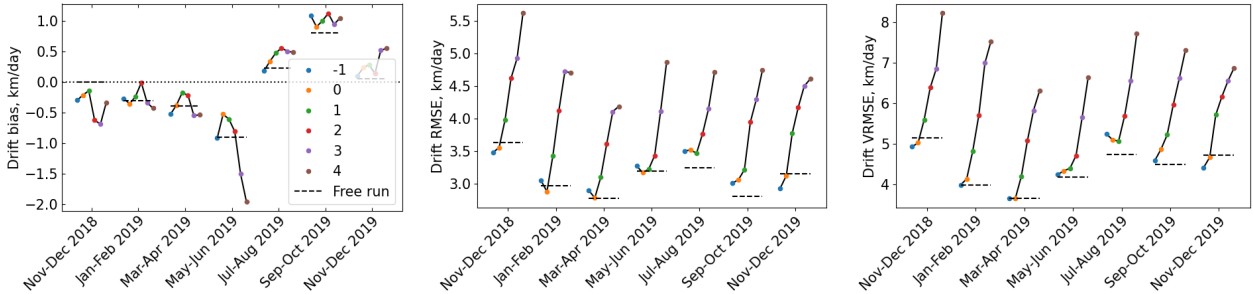

**Figure 12.** 2-monthly-averaged bias, RMSE and VRMSE for drift grouped by lead time (lead times are indicated in the figure legends). The lead times refer to the start of each 2-day drift evaluation period e.g. "0" covers the period from day 0 (12:00) to day 2 (12:00). The free run errors for the corresponding periods are plotted as dotted lines for reference. The bias is defined as modelled speed – observed speed.

Figure 9 shows the 2-monthly averages for the different metrics applied to the concentration and extent. We only plot results for the first year and two months, since the error statistics from the 2020 results were nearly identical to the previous years. As with the free run, the concentration from the forecast is generally higher than OSI-SAF in the winter months. As mentioned earlier, we don't try to correct this as reducing the concentration in the pack causes serious problem with the drift and thickness. The extent, which is adjusted during assimilation is also lower than the observed one for the whole period, by differing amounts. Figure 10 show some example forecasts from winter and autumn, one from January, March and September 2019 respectively. The first and third examples are forecasts with positive skill in extent (model does better than persistence), while the second has negative skill. However, they all have relatively low IIEE and RMSE in concentration. The January example shows that the rapid advance in the Kara Sea is captured relatively well by the model, and while the extent in the Greenland and Barents Seas is too high, the ice edge is quite variable and the model is doing better than the persistence at this time. The March example shows an example where there is quite a persistent negative bias to the northwest of Svalbard and in the Barents Sea. These errors probably originate in the ocean forcing and which we are unable to overcome, even with our flux compensation. The persistence forecast does well in this example since the ice conditions are not changing very much. In the September example, the bias is similar to the free run but it has been reduced by the assimilation. There is still a too-fast ice advance in the European Arctic but it is closer to the observations than the persistence forecast.

In the summer, The forecasts are systematically lower in concentration, partly due to differences in the pack, which we don't correct for, and in the extent. We are not too concerned by differences in the pack, as the differences are still comparable to observation error. Differences in the extent however are more concerning. In the summer, the forecasts score particularly badly with this metric, largely due to the dynamical problems also seen in the free run: arching above the Fram Strait, and fast ice in the Laptev and East Siberian seas that is too slow to detach. Figure 11 shows a typical example from this time period. Despite the correction in extent during assimilation the ice that is added in the Fram Strait is quite thin and it melts quite quickly.

Figure 12 shows the bias, RMSE and VRMSE for the forecast drift. For smaller lead times it has similar quality to the free run, which is to be expected since the assimilation does not modify fields in the pack, with minor differences that are probably

just due to differences in extent determining which observations are included or not. There is a small negative bias in speed in the winter months, and a slight positive bias in the summer. The bias is not showing a large variability with forecast lead time, although there are some exceptions. With the exception of the first two months, which is slightly higher than the others, the RMSE (RMSE in speed) is quite consistent throughout the whole evaluation period with a steady increase with lead time. The
RMSE for the final day is still mostly less than 5 km/day. The VRMSE, which evaluates the drift direction as well as its speed, shows a larger dependence on lead time, starting between $4.5-5$ km/day but getting up to between $7-8$ km/day by the final day (occasionally being under 7 km/day or over 8 km/day). This dependence on lead time is almost certainly due to increasing inaccuracy in the forecast winds.

## 5   Discussion and conclusions

neXtSIM-F became operational in July 2020 as part of CMEMS, and this paper presents an upgraded version that will be included in December 2020. Here we have evaluated the neXtSIM model itself in a free run for the period 1 November 2018 to 30 June 2020, as well as the neXtSIM-F forecast platform which corrects the initial conditions daily by assimilating sea ice concentration.

The free run, which used hindcast winds, had good drift, being relatively unbiased and having a low RMSE in speed of
$3-4$ km/day. The RMSE was closer to 2.5 km/day when less accurate observations (uncertainty less than 2.5 km for the 2-day drift) were filtered out, although the observations so removed would also be near the coast and in the MIZ where we do have problems with thickness and extent respectively. Considering the VRMSE, the RMSE in the final position of each trajectory, added about 1 km/day.

The model thickness was biased slightly too thick in general, and there was also a tendency to have a dipole behaviour with
one or too areas being too thick, and one or two being too thin. The location of these too-thin regions varied over the winter (the only time when the CS2-SMOS observations were available), but the thickness off northeast Greenland and north Svalbard was invariably too high. This caused the formation of an arch across the Fram Strait, which limited the export of ice through this strait in the summer, especially close to the coast.

When we evaluated the concentration and extent we found that the sea ice extent in the summer was greatly affected by the
arching mentioned above, in that the total extent was too small and that the ice was also displaced from where it should be — there was no ice close to the east coast of Greenland (unlike in the observations) and it extended too far to the east.

There were also less severe differences in the ice edge which were probably partly due to errors in the ocean and/or atmospheric forcings, but the other big issue was the extent of the landfast ice in May and June in the Laptev and East Siberian Seas — this was too slow to detach and also contributed to there being too little ice further away from the coast at this time.

Owing to the fairly conservative approach that we had to take to the assimilation the forecasts generally had the same properties as the free run with regards to thickness and drift. The bias in drift remains close to zero for the whole 8-day simulation, although the RMSE in speed and the VRMSE begin to deteriorate after about $4-5$ days of simulation (up to a lead time of 4) — this is almost certainly due to less accurate winds at these lead times. The forecasts also had greatly improved

IIEE scores compared to the free run. However, the more serious problems with the summer extent — in the Greenland Sea (due to arching) and to a lesser degree the Laptev and East Siberian seas (land fast ice) — were still present in the forecasts. The land-fast ice could possibly be improved with further tuning of the basal grounding scheme, while the arching problem is more difficult to solve. However, recent experiments indicate that changing the role of concentration in the rheology could give some improvements here — note that the viscous relaxation time in (??) and thus the entire left-hand side of (A2) only depend on damage and not on concentration so larger stresses do not drop very quickly if the ice is undamaged and the concentration drops (this makes the right-hand side of (A2) drop to near zero). There may be additional modifications to the rheology that could be made to help reduce arching in this region.

Assimilation of different data might also help:

1. Sea ice thickness: we could use either the hybrid CS2-SMOS product or the CS2 trajectories themselves to limit the build up of ice on either side of the Fram Strait. However, this data stops in mid-April so build-up that occurs later may still be enough to cause the arching.

2. Sea ice deformation: Korosov and Rampal (2017) were able to derive sea ice drift from SAR data using a combination of feature tracking and pattern matching. Deformation can then be calculated to modify the damage variable and thus induce break-up at the right place. Unfortunately, this option is also time-limited as surface melt makes the ice smoother and stops the feature tracking algorithm from working as effectively.

Another possibility could be to take the approach of Ying (2019) and to morph the modelled ice mask onto the observed one so that, for example, ice in the Greenland sea could be moved over towards the coast, instead of removing thicker ice at the ice edge and adding thinner ice towards the coast. Possibly a better source of ice extent than OSI-SAF concentration could also be used in conjunction with such a morphing approach. For example, the United States Naval Ice Center[6] produces daily, pan-Arctic ice charts. Other alternatives for assimilating extent are MASIE (Multisensor Analyzed Sea Ice Extent: Fetterer et al., 2010) and IMS (Interactive Multisensor Snow and Ice Mapping System: Helfrich et al., 2007), as done by the GOFS 3.1 system for example. Automatic ice type classification from SAR is also possible now e.g. the algorithm developed by Park et al. (2019) and upgraded by Boulze et al. (2020) will become operationally distributed by CMEMS in 2021.

A framework to produce an ensemble forecast with neXtSIM-F is also being developed (Cheng et al., submitted) (as a follow-up of the work of Rabatel et al., 2018), with the ultimate aim of using the Ensemble Kalman Filter (EnKF) assimilation method. Work on using EnKF with models running on adaptive meshes (like neXtSIM) is being developed in parallel at NERSC (Aydoğdu et al., 2019). This may also be effective at addressing some of the errors in the forecast.

Finally, in order to provide more consistent ocean inputs to the sea ice model (as well as to provide more realistic stresses and fluxes to the ocean model), work is ongoing to couple neXtSIM with the ocean models HYCOM and NEMO, and to add an atmospheric boundary layer model to mediate between the atmospheric model and the ice and/or ocean models. Also, neXtSIM is already coupled to the WAVEWATCH 3 wave model (Boutin et al., 2020), so there is scope for neXtSIM-F to include more components like wave and ocean models.

---

[6]https://www.natice.noaa.gov/Main_Products.htm

## Appendix A: Summary of Brittle Bingham-Maxwell (BBM) rheological equations

We refer readers to Rampal et al. (2019) for most of the details but give some important modifications below. The momentum balance is

$$\rho_i(h + h_y)\frac{D\mathbf{u}}{Dt} = \nabla \cdot (h\boldsymbol{\sigma} + \boldsymbol{\sigma}_P) + \tau - \rho_i(h + h_y)(f\mathbf{k} \times \mathbf{u} + g\nabla\eta), \tag{A1}$$

where $\mathbf{u} = (u, v)$ is the ice velocity, $\rho_i$ is the density of ice, $h_y$ is the mean young ice thickness, $h$ is the mean thickness of older ice, $\boldsymbol{\sigma}$ is the internal stress, $\boldsymbol{\sigma}_P$ is an additional plastic stress that is most active when damaged ice is under convergence, $\tau$ is the sum of stresses applied by the wind and ocean currents and by keels grounding on the sea floor, $f$ is the Coriolis parameter, $\mathbf{k} \times \mathbf{u} = (v, -u)$, $g$ is the acceleration due to gravity and $\eta$ is the sea surface height.

There is an evolution equation for the damage ($d$), which is unchanged from (Rampal et al., 2019, see equations A8 – A15). The rheology consists of a friction element and a dashpot in parallel, and a spring in series with this parallel element. Accordingly the stress follows the constitutive relation

$$\dot{\boldsymbol{\sigma}} + \frac{\boldsymbol{\sigma}}{\lambda}\left(\widetilde{P} + \frac{\lambda\dot{d}}{1-d}\right) = E\mathbf{K} : \dot{\varepsilon}, \tag{A2}$$

where $\mathbf{K}$ is the stiffness tensor (see equation A4 of Rampal et al., 2019), $\dot{\varepsilon}$ is the deformation rate tensor, $E$ is the Young's modulus and $\lambda$ is the viscous relaxation time. $E$ and $\lambda$ vary according to the damage and the concentration of old ice $c$

$$E = e^{-0.2(100-c)}(1-d)E_0, \tag{A3a}$$

$$\lambda = \lambda_0(1-d)^{\alpha-1}. \tag{A3b}$$

Here $E_0$ is the Young's modulus for undamaged ice at 100% concentration, $\lambda_0$ is the (large) value of $\lambda$ for undamaged ice, and $\alpha > 1$ is an exponent which can be tuned to control how fast $\lambda$ drops due to damaging. Note we don't consider young ice in the constitutive relations since it is assumed to be weak. The term $\widetilde{P}$ depends on the normal stress $\sigma_n = \frac{1}{2}(\sigma_{11} + \sigma_{22})$ (defined to be positive under convergent conditions), and particularly on whether the ice is converging or not:

$$\widetilde{P} = \begin{cases} 1 & \text{for } \sigma_n \leq 0 \text{ (diverging)}, \\ 0 & \text{for } 0 < \sigma_n < P_{\max} \text{ (converging and elastic)}, \\ 1 - \frac{P_{\max}}{\sigma_n} & \text{for } \sigma_n > P_{\max} \text{ (converging and ridging)}, \end{cases} \tag{A4}$$

where $P_{\max} = P_* h^2 e^{-0.2(100-c)}$ is the concentration- and thickness-dependent threshold for when the Bingham element starts moving ($P_*$ is a constant tuning parameter of the order of $10\,\text{kPa}$). If $\sigma_n < 0$, the friction element is inactive and the dashpot is free to move, and the internal stress is quickly dissipated in damaged areas. If $0 < \sigma_n < P_{\max}$, the friction element is active and stationary, which also stops the dashpot from moving and dissipating stress. Thus even highly damaged ice is able to resist compression. However, once $\sigma_n > P_{\max}$, the friction element starts to move, the dashpot becomes active again, and the ice starts to ridge.

The MEB rheology as presented by Rampal et al. (2019) corresponds to the situation where $\widetilde{P}$ is always 1 in (A4). In addition the term involving $\dot{d}$ in (A2) was not present (it was required to stabilise the explicit solution of the BBM equations). In the first version of our CMEMS forecast (which ran from Jul – Dec 2020), two modifications were made to the version of MEB in Rampal et al. (2019):

(i) In the equation (A3a) for $E$, $d$ was taken to be zero if it was less than $0.95$, which helped to improve localisation of deformation.

     (ii) An extra stress $\boldsymbol{\sigma}_{\mathrm{P}}$ was added to $\boldsymbol{\sigma}$ in the momentum equation (A1) which provided some resistance to compression. This stress was given by

$$\boldsymbol{\sigma}_{\mathrm{P}} = \frac{h^2 \mathrm{e}^{-0.2(100-c)} P_{**}}{|\nabla \cdot \mathbf{u}| + \delta_{\mathrm{P}}} \begin{pmatrix} 1 & 0 & 0 \\ 0 & 1 & 0 \\ 0 & 0 & \frac{1}{2} \end{pmatrix} \begin{pmatrix} \dot{\varepsilon}_{11} \\ \dot{\varepsilon}_{22} \\ \dot{\varepsilon}_{12} \end{pmatrix}, \tag{A5}$$

where $P_{**}$ was a tuning parameter of the order of $10\,\mathrm{kPa}$, and $\delta_{\mathrm{P}}$ was a small parameter needed for numerical stability.

*Author contributions.* TW and AK wrote the paper and did the experiments. PR and EO contributed to writing and planning the paper and experiments. EO was also behind many background developments in the model code.

*Competing interests.* No competing interests.

*Acknowledgements.* We would like to acknowledge support from the Research Council of Norway (RCN) projects neXtWIM #244001 and
Nansen Legacy #276730. The projects ImpSIM 18CP10 funded by the French Navy (SHOM) and Cryosphere Virtual Laboratory (CVL, 4000128808/19/I-NS) funded by the European Space Agency also supported the last phase of this manuscript preparation. We also benefited from work done during the project SWARP funded by the European Commission FP7 programme (grant number 607476); from interesting discussions with Véronique Dansereau on the MEB and BBM rheology; from the work of Abdoulaye Samaké who parallelised the neXtSIM code; and from the contribution of Mika Malila to the evaluation scripts. Finally, we would like to thank Sylvain Bouillon and Philip Griewank
for their original contribution to building a forecasting system around the neXtSIM sea ice model.

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
