# Peer review of "Presentation and evaluation of the Arctic sea ice forecasting system neXtSIM-F"

_The Cryosphere, 2019_

## Referee Comment (RC1) · J.-F. Lemieux (Referee) · 23 Aug 2019

Review of **Presentation and evaluation of the Arctic sea ice forecasting system neXtSIM-F** by Williams et al.

In this paper the authors present and evaluate a new sea ice forecasting system based on the neXtSIM sea ice model. They have evaluated drift, sea ice thickness and concentration against different datasets and show that the forecasts, in general, show good agreement with the observations. Being also in the business of ice-ocean predictions, I recognize the amount of work that was done by the authors to develop and evaluate this kind of forecasting system.

[Figure]

I have, however, two major concerns about the paper.

The first one is easy to address. I think the authors need to do a better literature review. We have the impression there are currently only two sea ice forecasting systems (Topaz and GOFS) in the world and that neXtSIM-F is the third one to be proposed. For example, the Danish meteorological institute, the UK Met office, ECMWF, Environment and Climate Change Canada (ECCC) all have operational sea ice forecasting systems. Note also that we (ECCC) have, a few years ago, developed a similar system as neXtSIM-F: a stand alone sea ice model coupled to a slab ocean model (with MLD and SSS initialized from a coupled ice-ocean prediction system and SST from an analysis...). It is not used anymore as it has been replaced by a coupled ice-ocean forecasting system but I think, given the similarities between the two systems, that it should be mentioned. You could then describe what is different (e.g. the ocean heat flux correction you propose...which is interesting by the way.)

The second major concern I have is that neXtSIM-F was evaluated for only half a year. And especially, the evaluation was done during the winter months, that is at a time when there is almost no navigation in the Arctic. I would understand why you would choose this period if your system was designed specifically for the Baltic Sea or the Gulf of St-Lawrence (where there is a lot of navigation during winter). I have the impression that the authors have submitted their paper too early and that it would make more sense to show a complete seasonal cycle of the forecast scores. You mention, anyway, that neXtSIM-F will be operational in November of this year...this is coming soon. This means you will soon have the evaluation for the summer months? Then I really think you should include these.

[Figure]

This paper will be an important contribution to the field of sea ice forecasting but first the authors need to address these concerns and the minor comments given below.

**1   Minor comments**

1) p.5 line 19. I guess you use a turning angle for the ocean-ice stress? Please mention it.

2) p.5 line 22. this is already mentioned above.

3) sections 3.6-3.7. What is this the atmospheric forcing you will use once the system becomes operational? I hope it will be the same one used for the evaluation because then it does not make sense.

4) p.8 lines 7-9. You give information that is not needed. We don't need to know that it was first coded in Matlab. Just say that the version of neXtSIM used for neXtSIM-F is described in Rampal et al. 2016 and Samake et al 2017.

5) p.8 lines 13-14. Mention that the MLD varies spatially and that this spatial field is fixed (I guess) during the 7 days of the forecast.

6) section 4.2. Mention how you initialize the sea ice velocity.

7) section 4.2. What justifies the values of 0.2 and 0.8 for the initial thin and thick ice?
8) p.10 line 19. The first condition (i.e. $c_t > 0$) is not needed, right?

9) section 5.1. Clearly state how long (time period) is the free run.

10) p. 11, line 6. Typo "the no ice".

11) Fig. 3. Explain how you calculate the concentration and uncertainty. Is it the mean over the whole domain? Then it means you have many grid cells with a concentration of zero and many with a concentration close to 1.0. This means the signal you are interested in is kind of buried because most grid cells have a forecast concentration close to the observed one...which is not surprising. How do you deal with uncertainties when the concentration is 1 or 0...it cannot be gaussian, right?

12) p. 13, lines 6-7 and Fig. 6. Is it possible the MEB rheology leads to too much convergence? (e.g. North of the CAA).

13) p. 13, lines 18-24. Too many numbers given. I don't think you need to give all these values.

14) p. 13, line 27. What do you mean by "eroded"? Please clarify.

15) p. 13, line 33. I don't think the ice is landfast there but it is (very) slowly drifting.

16) p. 14, line 1. remove "very respectable"...just give the number and that's it.

17) p. 14, line 10-14. Please clarify this paragraph. It is not clear here what are exactly the experiments (especially the sentence"...without assimilation, with assimilation of concentration and with assimilation of concentration and thickness.").

18) Fig. 9. I am not surprised persistence is doing so bad here because the beginning of November is a time when there is a lot of ice growth. No wonder the model performs so well. I think it would be good to show another case for example in March. Is it always true that the model beats persistence? Is is always true that assimilation improves the quality of the forecast?

19) p. 16, lines 13-14: The captions in Fig. 10 and 11 do not match the text here about the dates.

20) p. 18, line 1: East Siberian and Chukchi seas...not obvious to me when I look at Figs 10 and 11.

21) Fig. 13 and 14 are not discussed. Are they really needed?

22) p. 19, lines 11-13: Please rephrase...

23) p. 20, line 1. What drifters are we talking about here?

24) p. 25, lines 11-17. I think it could also be the thermodynamic model itself. There is clearly too much ice growth...the model might require some tuning. This should be mentioned. What about the way the thin and thick ice categories are initialized? How

Interactive
comment

does it affect the overestimation of the growth?

25) p. 25, line 19. What do you mean by "fragmentation"?

26) p. 25, line 23. I thought neXtSIM is using our grounding scheme for landfast ice? Would it be worth tuning the k1 parameter?

27) p. 26, lines 20-22. Is it a result you presented in this paper? Or I just don't understand the sentence. Please clarify this.

Congratulations for your work on developing this new sea ice forecasting system.

Jean-François Lemieux

————————————————————————

---

## Referee Comment (RC2) · Anonymous Referee #2 · 30 Aug 2019

In this paper the authors introduce a new sea ice forecasting system neXtSIM-F based on the neXtSIM sea ice model and present an evaluation of the model over a single season - winter 2018-19.

I feel that this study will be worth publishing in The Cryosphere (although it would likely fit better in GMD than TC). However several changes will be required before this is possible.

*General comments*

1. It is not made clear enough what the various runs and systems are that are assessed in this study. In particular there is also no mention of the "free run" before it is evaluated in Section 5.1. Section 3 contains information on the observational datasets used

in this study but there is no equivalent for the model datasets. This study needs a summary of exactly which runs and systems are being evaluated with perhaps a table.

2. Additionally the names neXtSIM and neXtSIM-F seem to be fairly inter-changeable in the manuscript. I guess the neXtSIM-F forecasting system uses the neXtSIM sea ice model. If so then I think the name of your forecasting system as neXtSIM-F is a bit confusing.

3. The evaluation period is only a few months and does not include the late spring/summer period when many sea ice forecasting systems report their poorest performance. This means that it is hard to put the evaluation here into context with other operational systems. The conclusions of this study would be much strengthened if the authors could perform, and evaluate, a complete annual cycle (or preferably 2).

4. In general I find that there are too many figures and stats in the paper, which makes it hard to understand what the take-home message is.

*Specific comments*

1. In many places the language used in the paper is too informal and colloquial (i.e., in the abstract we have "...in our system, we obtain..." and on P9 we have "the observed ones").

2. You need to be careful to distinguish between "sea ice concentration", which ranges from 0 to 100%, and "sea ice area fraction", which ranges from 0 to 1 throughout this manuscript. For example in Figure 4 the caption says "concentration" but the scale is +/- 0.5%. This is either a low "concentration" or a high "area fraction". I assumed the former to start with until I noticed that the text talks about an associated reduction in extent. With changes of +/- 0.5% concentration I wouldn't expect to see any departure to the 15% contour (extent) so is it actually "area fraction" plotted here?

3. I find the abstract to be rather technical and not very abstract. It reads a bit more like a conclusions/summary section. I would encourage the authors to make the abstract

more exciting to make the paper more inviting to potential readers.

4. The introduction section (section #1) is rather disjointed. It starts with some motivation for sea ice forecasting (with background on changing climate) but then jumps straight in to say that neXtSIM-F is based on neXtSIM. It doesn't actually say that neXtSIM(-F) is a sea ice forecasting system! It would be better to include a couple of extra lines to say that this is the case. Perhaps to say something like "Here we introduce a new sea ice forecasting system, neXtSIM-F, that is based upon the neXtSIM model...".

5. I find the introduction to operational ocean forecasting systems in Section 2 to be, almost paradoxically, both too detailed and non-existent. I say too detailed because I am left wondering why there is such a thorough introduction provided to the GOFS system when it isn't really used in this study? Of course, GOFS is only one of many global operational ocean-sea ice forecasting systems and you don't mention any others apart from TOPAZ and neXtSIM. The Tonani et al. (2015) GODAE paper provides a nice reference describing the world's operational global forecasting systems. Although several of the systems have doubtlessly moved on since 2015, this reference provides evidence for the breadth of activity in the world. Tonani, M., Balmaseda, M., Bertino, L., Blockley, E. W., Brassington, G., Davidson, F., Drillet, Y., Hogan, P., Kuragano, T., Lee, T., Mehra, A., Paranathara, F., Tanajura, C. A. S. and Wang, H.: Status and future of global and regional ocean prediction systems, J. Oper. Oceanogr., 8, sup2, s201-s220, doi:10.1080/1755876X.2015.1049892, 2015.

7. The data sources section (#3) does not make it clear which datasets are used for assimilation and which are used for evaluation (and hence which are used for both). At the least it is important to note which datasets are independent from the assimilation.

8. Related to the above point I find the description of the blended SSMIS+AMSR2 product somewhat confusing. Is this done purely for the evaluation? If not why can't the DA do this blending by waiting the observations with their respective errors?

9. P3, L4: "...profiles from Argo floats.". Why do you only use Argo floats (if that's true)? Why not CTD/XBT/seals etc.?

10. I do not understand why a couple of weeks of CFSv2 is used in place or ECMWF. Surely you could get the replacement data from somewhere else (like ECMWF themselves for example)? If not then you should really consider the implications of using CFSv2. Specifically: is this the configuration with unrealistic ice growth caused by the fact that they turned off the stratus cloud formation to improve tropical temperatures and ENSO predictability (as described by Yang et al. 2017 and references therein)?: Yang, Q., M. Wang, J.E. Overland, W. Wang, and T.W. Collow, 2017: Impact of Model Physics on Seasonal Forecasts of Surface Air Temperature in the Arctic. Mon. Wea. Rev., 145, 773–782, https://doi.org/10.1175/MWR-D-16-0272.1

11. I don't like your "RMSE" for extent as it is exactly the Integrated Ice Edge Error (IIEE) of Goessling et al., (2016). You cite the ensemble extension of the IIEE (the SPS paper of Goessling and Jung, 2018) and say that your RMSE is like a deterministic version of that, which is misleading. It would be better to just cite the 2016 paper instead and call your metric "IIEE" instead of "RMSE": Goessling, H. F., Tietsche, S., Day, J. J., Hawkins, E., and Jung, T.: Predictability of the Arctic sea ice edge, Geophys. Res. Lett., 43, 1642–1650, https://doi.org/10.1002/2015GL067232, 2016

12. In Figure 3 I note that the neXtSIM concentration evolution is very smooth – more so than the low resolution SSMIS data – which I didn't expect given the resolution of the model. Can you comment on this? Is this caused by the fact that neXtSIM still uses the continuum formulation and so doesn't resolve small scale features?

13. I note with interest that MOSAiC forecasting is mentioned as a motivation for improving sea ice forecasts. There is an international project (SIDFEx) currently coordinating operational sea ice drift forecasts specifically to provide guidance to the Polarstern/MOSAiC. Presently the list of models includes TOPAZ but not neXtSIM. Are there any plans to contribute neXtSIM drift forecasts to SIDFEx? This might be an

interesting way to show the skill of neXtSIM in this regard.

14. Some of the figures (e.g., Figs 10&11) suggest that the data assimilation is having a rather modest impact on the forecasts compared with many of the operational systems that I have seen in the past. Can you comment as to why that might be?

15. On page 20 it is mentioned that the "RMSE for drifters placed on the first day..." but this is the 1st time in the manuscript that drifters are mentioned. Can you explain this a bit more please?

*Figures*

As mentioned above I feel that there are too many figures in this manuscript. In particular in Figs 11 & 12 there are 12 panels and each row looks virtually identical. Apart from telling me that the assimilation is having a rather modest impact, I don't understand what I'm supposed to do with all this information. Additionally the next similar set of figures, Figs 13-14, don't even seem to be discussed in the text at all. So are they necessary?

Many of the figure captions are too brief and should be improved. I believe that the Copernicus journal guidelines are that figures should be able to work stand-alone from the text, for which a bit more information is required.

I find that the x-axis date tick-marks provided on the time-series plots (Figs 3, 5, 7, 12 – less so for Fig 9) are not very useful. With such a short run period it would be better to include more dates. At the very least there minor tick marks should be used to show each day (or 5-days or something). It would also be good to specify this in the figure caption perhaps.

Figure 1 is a bit confusing because I am left wondering whether different time-scales are involved here. Is this a daily schematic or does it depict the whole run? For example the 2 top boxes (initialization) are surely not done each day are they? If not then it should be made clear what is done each day and what isn't – either in the caption or

the figure (or both). Perhaps the initialization boxes could be enclosed in a dotted box or something?

I suggest you should also re-think your use of red-blue colour maps for sea ice concentration. I have seen people use red for less ice (as it's hotter) and blue for more ice (colder) in the past as well as the other way around. It might be better to avoid the use of a hot<->cold colour-map therefore.

*Technical corrections*

The 1st instance of "SSMIS" is correct but thereafter it has been changed to "SMMIS".

Also "first day (4th day)" appears in many places, which is not very consistent

P2, L26: CMEMS should be "Copernicus Marine Environment Monitoring Service"

P2, L29: "...the version 4.1 of the..." – suggest to remove the 1st instance of "the" here

P2, L30: The reference for CICE v4.1 is Hunke and Lipscomb (2010): Hunke, E. C. and Lipscomb, W. H.: CICE: the Los Alamos sea ice model. Documentation and software users manual, Version 4.1 (LA-CC-06-012), T-3 Fluid Dynamics Group, Los Alamos National Laboratory, Los Alamos, 2010

P3, L30: "As specified by the validation reports above..." should be "As specified by the validation reports cited above..."

P4, L5: "metrics" should be "metric"

P4, L6: is extent "above 15%" or "at least 15%"? I thought the latter.

P4, L13: "...can be obtained for 48 hours..." sounds like only 48 hours of data. Do you mean this or do you mean the data is available 48-hours behind real-time?

P5, L5: calculating volume for each model & obs based on thickness like this will involve different areas of ice won't it?

P5, L26: "Modelling" is spelt incorrectly as "Modeling" in the NEMO acronym

P5, L31: ". . .if the temperature is below 0C". Which temperature – surface skin temperature, or near-surface atmosphere temperature (T2M)? Please be more specific.

P10, L29: "of these variables" is not adding to this sentence and should be removed

P11, L6: ". . .predicts the no ice. . .". Please remove the "the".

P11, L11: "averages values" should be "average values"

P11, L19: "underestimation in the Bering Sea". I presume that you mean the Chukchi Sea here because the Bering Sea is outside your model domain?

P13, L33-4: "land-fast" should be "land-fast ice"?

P13, L35: (as above) I suspect that "Bering Sea" should be "Chukchi Sea" here

P14, L9: the title "Evaluation of forecasts with assimilation" is confusing because I doubt that you are actually doing assimilation in your forecasts are you? Perhaps this should be changed to something more like "Forecasts performed from analysed ice conditions"?

P16, L9: Do you mean "significantly" here in the scientific sense of the word? If so include a p-value, if not I suggest changing to "considerably".

P18, Fig 9 caption: "(blue)", "(orange)", and "(red)" are provided but not "(green)"

P21, Fig 12 caption: "error bars" should be "shading"

P26, L7: "limited resources" suggests a deficiency in resource. You should change to "minimal resources" if you wish to suggest that the model is cheap to run.

P26, L20-21: ". . .forecasts used saved atmospheric and ocean forecasts as forcing. . .". What does this mean?

---

## Author Comment (AC1) · 7 Nov 2020

Response to reviewer 1 (JF Lemieux)

Dear Jean-François,
Thank you very much for your helpful comments. We will respond to them below.

***Review of Presentation and evaluation of the Arctic sea ice forecasting system neXtSIM-F by Williams et al.***
*In this paper the authors present and evaluate a new sea ice forecasting system based on the neXtSIM sea ice model. They have evaluated drift, sea ice thickness and concentration against different datasets and show that the forecasts, in general, show good agreement with the observations. Being also in the business of ice-ocean predictions, I recognize the amount of work that was done by the authors to develop and evaluate this kind of forecasting system.*

*I have, however, two major concerns about the paper.*
*The first one is easy to address. I think the authors need to do a better literature review. We have the impression there are currently only two sea ice forecasting systems (Topaz and GOFS) in the world and that neXtSIM-F is the third one to be proposed. For example, the Danish meteorological institute, the UK Met office, ECMWF, Environment and Climate Change Canada (ECCC) all have operational sea ice forecasting systems. Note also that we (ECCC) have, a few years ago, developed a similar system as neXtSIM-F: a stand alone sea ice model coupled to a slab ocean model (with MLD and SSS initialized from a coupled ice-ocean prediction system and SST from an analysis...). It is not used anymore as it has been replaced by a coupled ice-ocean forecasting system but I think, given the similarities between the two systems, that it should be mentioned. You could then describe what is different (e.g. the ocean heat flux correction you propose...which is interesting by the way.)*
We agree that our discussion gives the wrong impression of the field, and are quite sorry about this. We have now scaled back our discussion to refer to the papers of Tonani et al (2015), and also the more recent one by Hunke et al (2020, https://link.springer.com/article/10.1007/s40641-020-00162-y) which also gives an idea about trends in the systems. Thanks for the reference about the ECCC stand-alone sea ice forecast RIPS that you sent - we have included it in our introduction as well.

*The second major concern I have is that neXtSIM-F was evaluated for only half a year. And especially, the evaluation was done during the winter months, that is at a time when there is almost no navigation in the Arctic. I would understand why you would choose this period if your system was designed specifically for the Baltic Sea or the Gulf of St-Lawrence (where there is a lot of navigation during winter). I have the impression that the authors have submitted their paper too early and that it would make more sense to show a complete seasonal cycle of the forecast scores. You mention, anyway, that neXtSIM-F will be operational in November of this year...this is coming soon. This means you will soon have the evaluation for the summer months? Then I really think you should include these.*

We agree that this was a major short-coming of the previous version of the paper, and have now completed evaluation for a period of 20 months. neXtSIM-F finally went operational in July 2020, and will be updated in December with the version evaluated in the present paper.

*This paper will be an important contribution to the field of sea ice forecasting but*
*first the authors need to address these concerns and the minor comments given below.*
*1 Minor comments*
*1) p.5 line 19. I guess you use a turning angle for the ocean-ice stress? Please*
*mention it.*

Yes, it is 25 degrees. Added to paper (section 4.2)

*2) p.5 line 22. this is already mentioned above.*

We have simplified this section and added it to section 4.1, where the slab ocean is introduced.

*3) sections 3.6-3.7. What is this the atmospheric forcing you will use once the system*
*becomes operational? I hope it will be the same one used for the evaluation because*
*then it does not make sense.*

The forcing we will use is the ECMWF forecast described in section 3.6. A line has been added to clarify this.

*4) p.8 lines 7-9. You give information that is not needed. We don't need to know that it*
*was first coded in Matlab. Just say that the version of neXtSIM used for neXtSIM-F is*
*described in Rampal et al. 2016 and Samake et al 2017.*

OK - we have removed some of the history.

*5) p.8 lines 13-14. Mention that the MLD varies spatially and that this spatial field is*
*fixed (I guess) during the 7 days of the forecast.*

It actually varies with time according to the ocean forecast. We have added a comment to clarify this.

*6) section 4.2. Mention how you initialize the sea ice velocity.*

We added a sentence about this: the ice velocity and also the damage are started from zero.

*7) section 4.2. What justifies the values of 0.2 and 0.8 for the initial thin and thick ice?*

This is a little arbitrary (necessarily so given the difficulty of determining the concentration of thin ice) but the model wasn't too sensitive to this.

*8) p.10 line 19. The first condition (i.e. $c_t > 0$) is not needed, right?*

Yes you are right - we have removed it.

*9) section 5.1. Clearly state how long (time period) is the free run.*
*10) p. 11, line 6. Typo "the no ice".*

Fixed.

*11) Fig. 3. Explain how you calculate the concentration and uncertainty. Is it the mean*
*over the whole domain? Then it means you have many grid cells with a concentration*
*of zero and many with a concentration close to 1.0. This means the signal you are*
*interested in is kind of buried because most grid cells have a forecast concentration*
*close to the observed one...which is not surprising.*

We actually average over the union of the ice masks. This still leaves the pack ice, which in our case is usually close to 1, while the observations is usually about 0.9-1. So this signal is less interesting since the observations are also uncertain. The extent is actually the main signal we would like to capture, but we present the comparison of the concentration also.

*How do you deal with uncertainties when the concentration is 1 or 0...it cannot be gaussian, right?*

This is a good question. It turns out that it was quite difficult to estimate uncertainties since we did not know the distribution of the observation errors. Therefore we removed the error shading from most of the plots. For the error plots we kept it as a rough reference level to show when our results were significantly different from the observations. Another interesting thing that came up from the uncertainty analysis was with the drift speed - adding noise to a given set of "true" drift values introduced a bias of $2\sigma^2$ to $<drift^2>$ where $\sigma^2$ is the variance of the drift. Hence one should be a bit careful when comparing drift speeds.

*12) p. 13, lines 6-7 and Fig. 6. Is it possible the MEB rheology leads to too much convergence? (e.g. North of the CAA).*

Yes. It is slightly improved with BBM, but there are still issues with this near the northeast coast of Greenland and the north coast of Svalbard. This has some unfortunate follow-on effects on the summer ice extent in the Greenland Sea especially.

*13) p. 13, lines 18-24. Too many numbers given. I don't think you need to give all these values.*

We have reduced the number of numbers inline throughout.

*14) p. 13, line 27. What do you mean by "eroded"? Please clarify.*

We mean that by setting a maximum threshold on the uncertainty in the ice drift we had masked the observations in some areas (mainly near the ice edge, coast, and the north pole). We now generally use a higher threshold (10 km/day) so that we can include evaluation in summer, but include a table where we use the lower threshold (1.25 km/day) for 2 winters and show it reduces our errors significantly. However some of the regions with higher errors (eg coast and MIZ) are regions where we have some problems with other variables like thickness and extent so masking them out can help for that reason too.

*15) p. 13, line 33. I don't think the ice is landfast there but it is (very) slowly drifting.*

Yes, you are right.

*16) p. 14, line 1. remove "very respectable"...just give the number and that's it.*

Fixed.

*17) p. 14, line 10-14. Please clarify this paragraph. It is not clear here what are exactly the experiments (especially the sentence"...without assimilation, with assimilation of concentration and with assimilation of concentration and thickness.").*

We have now reduced the number of experiments to one free run and one with assimilation of concentration/extent.

*18) Fig. 9. I am not surprised persistence is doing so bad here because the beginning of November is a time when there is a lot of ice growth. No wonder the model performs so well. I think it would be good to show another case for example in March. Is it always true that the model beats persistence? Is is always true that assimilation improves the quality of the forecast?*

Assimilation always improves the forecast concentration and extent compared to the free run (see the new figure 9). However it only beats the persistence in extent in the months Sep-Feb. We now have 4 examples (2 with positive skill (Jan, Sep) and 2 with negative skill (Mar, Jun)).

*19) p. 16, lines 13-14: The captions in Fig. 10 and 11 do not match the text here about the dates.*

Figs 10 and 11 have now been removed.

*20) p. 18, line 1: East Siberian and Chukchi seas...not obvious to me when I look at Figs 10 and 11.*

These figures and the discussion have now been replaced.

*21) Fig. 13 and 14 are not discussed. Are they really needed?*

They have in fact been replaced.

*22) p. 19, lines 11-13: Please rephrase…*

(This referred to the evaluation against SMOS thin ice thickness.) We decided to remove evaluation against SMOS since errors were dominated by difference in extent.

*23) p. 20, line 1. What drifters are we talking about here?*

Every day at 12:00 we place Lagrangian drifters at the grid points of the OSISAF drift product and advect them for 48h at the ice velocity. The total drift is then compared to the OSISAF drift product. We clarify this in the Data Sources section.

*24) p. 25, lines 11-17. I think it could also be the thermodynamic model itself. There is clearly too much ice growth...the model might require some tuning. This should be mentioned. What about the way the thin and thick ice categories are initialized? How does it affect the overestimation of the growth?*

We have now removed the comparison to SMOS.

*25) p. 25, line 19. What do you mean by "fragmentation"?*

Technically we mean damaging. We have changed this in the paper.

*26) p. 25, line 23. I thought neXtSIM is using our grounding scheme for landfast ice? Would it be worth tuning the k1 parameter?*

We now have the opposite problem of too much fast ice in the Laptev and East Siberian sea. We are currently tuning this parameter, but for this paper we were concentrating on the main parameters of the dynamics (cohesion, Pmax, drag)

*27) p. 26, lines 20-22. Is it a result you presented in this paper? Or I just don't understand the sentence. Please clarify this.*

Yes it is presented. We clarify the different runs better now (eg free run v forecasts)

*Congratulations for your work on developing this new sea ice forecasting system.*

*Jean-François Lemieux*

Thank you.

---

## Author Comment (AC2) · 7 Nov 2020

Dear anonymous reviewer, thank you for your thorough review and helpful comments. We will respond to them below.

*In this paper the authors introduce a new sea ice forecasting system neXtSIM-F based on the neXtSIM sea ice model and present an evaluation of the model over a single season - winter 2018-19.*
*I feel that this study will be worth publishing in The Cryosphere (although it would likely fit better in GMD than TC). However several changes will be required before this is possible.*

**General comments**

*1. It is not made clear enough what the various runs and systems are that are assessed in this study. In particular there is also no mention of the "free run" before it is evaluated in Section 5.1. Section 3 contains information on the observational datasets used in this study but there is no equivalent for the model datasets. This study needs a summary of exactly which runs and systems are being evaluated with perhaps a table.*
This has now been clarified in a few places  (introduction, start of results section, discussion and conclusions).
*2. Additionally the names neXtSIM and neXtSIM-F seem to be fairly inter-changeable in the manuscript. I guess the neXtSIM-F forecasting system uses the neXtSIM sea ice model. If so then I think the name of your forecasting system as neXtSIM-F is a bit confusing.*
This has now been clarified in a few places.
*3. The evaluation period is only a few months and does not include the late spring/summer period when many sea ice forecasting systems report their poorest performance. This means that it is hard to put the evaluation here into context with other operational systems. The conclusions of this study would be much strengthened if the authors could perform, and evaluate, a complete annual cycle (or preferably 2).*

This is a completely valid criticism and the run is now 20 months. We also have our worst performance in the summer months.

*4. In general I find that there are too many figures and stats in the paper, which makes it hard to understand what the take-home message is.*

We have removed some figures and tables, and replaced some others, and also tried to emphasise the main take-home message more, so hopefully the new figures and tables now help understanding rather than hinder it.

**Specific comments**

*1. In many places the language used in the paper is too informal and colloquial (i.e., in the abstract we have ". . .in our system, we obtain. . ." and on P9 we have "the observed ones").*

We have tried to make the language more formal.

*2. You need to be careful to distinguish between "sea ice concentration", which ranges from 0 to 100%, and "sea ice area fraction", which ranges from 0 to 1 throughout this manuscript. For example in Figure 4 the caption says "concentration" but the scale is +/- 0.5%. This is either a low "concentration" or a high "area fraction". I assumed the former to start with until I noticed that the text talks about an associated reduction in extent. With changes of +/- 0.5% concentration I wouldn't expect to see any departure to the 15% contour (extent) so is it actually "area fraction" plotted here?*

You are correct that it was actually area fraction - we have changed to using concentration with units of % everywhere.

*3. I find the abstract to be rather technical and not very abstract. It reads a bit more like a conclusions/summary section. I would encourage the authors to make the abstract more exciting to make the paper more inviting to potential readers.*

We have removed the numbers from the abstract and tried to make it more inviting.

*4. The introduction section (section #1) is rather disjointed. It starts with some motivation for sea ice forecasting (with background on changing climate) but then jumps straight in to say that neXtSIM-F is based on neXtSIM. It doesn't actually say that neXtSIM(-F) is a sea ice forecasting system! It would be better to include a couple of extra lines to say that this is the case. Perhaps to say something like "Here we introduce a new sea ice forecasting system, neXtSIM-F, that is based upon the neXtSIM model. . .".*

Thanks - this is a good sentence that we have used in the introduction. Hopefully introduction is now less disjointed.

*5. I find the introduction to operational ocean forecasting systems in Section 2 to be, almost paradoxically, both too detailed and non-existent. I say too detailed because I am left wondering why there is such a thorough introduction provided to the GOFS system when it isn't really used in this study? Of course, GOFS is only one of many global operational ocean-sea ice forecasting systems and you don't mention any others apart from TOPAZ and neXtSIM. The Tonani et al. (2015) GODAE paper provides a nice reference describing the world's operational global forecasting systems. Although several of the systems have doubtlessly moved on since 2015, this reference provides evidence for the breadth of activity in the world. Tonani, M., Balmaseda, M., Bertino, L.,*

*Blockley, E. W., Brassington, G., Davidson, F., Drillet, Y., Hogan, P., Kuragano, T., Lee, T., Mehra, A., Paranathara, F., Tanajura, C. A. S. and Wang, H.: Status and future of global and regional ocean prediction systems, J. Oper. Oceanogr., 8, sup2, s201-s220, doi:10.1080/1755876X.2015.1049892, 2015.*

We agree that our discussion gives the wrong impression of the field, and are quite sorry about this. We have now scaled back our discussion to refer to the papers of Tonani et al (2015), and also the more recent one by Hunke et al (2020, https://link.springer.com/article/10.1007/s40641-020-00162-y) which also gives an idea about trends in the systems.

*7. The data sources section (#3) does not make it clear which datasets are used for assimilation and which are used for evaluation (and hence which are used for both). At the least it is important to note which datasets are independent from the assimilation.*

A sentence has been added to the start of each data-source subsection to make it explicit which are used in assimilation and evaluation, and which are used in evaluation only.

*8. Related to the above point I find the description of the blended SSMIS+AMSR2 product somewhat confusing. Is this done purely for the evaluation? If not why can't the DA do this blending by waiting the observations with their respective errors?*

We now use this only for data assimilation, but it unfortunately became quite inconvenient to use it for evaluation due to missing sections of data in the AMSR2 product particularly. We clarify this in section 2.3.

*9. P3, L4: ". . .profiles from Argo floats.". Why do you only use Argo floats (if that's true)? Why not CTD/XBT/seals etc.?*

Here we were only describing what TOPAZ does. However, it also assimilates ice-tethered profiles as well. Other data like the ones you mention are available too late to assimilate in near-real-time. They are used in the TOPAZ reanalysis though (Laurent Bertino, *personal communication*).

*10. I do not understand why a couple of weeks of CFSv2 is used in place or ECMWF. Surely you could get the replacement data from somewhere else (like ECMWF them-selves for example)? If not then you should really consider the implications of using CFSv2. Specifically: is this the configuration with unrealistic ice growth caused by the fact that they turned off the stratus cloud formation to improve tropical temperatures and ENSO predictability (as described by Yang et al. 2017 and references therein)?: Yang, Q., M. Wang, J.E. Overland, W. Wang, and T.W. Collow, 2017: Impact of Model Physics on Seasonal Forecasts of Surface Air Temperature in the Arctic. Mon. Wea. Rev., 145, 773–782, https://doi.org/10.1175/MWR-D-16-0272.1*

We have now been able to access those missing forecasts and have rerun the free run and the relevant forecasts without needing to plug with CFSv2. This also allows us to delete the section describing CFSv2.

*11. I don't like your "RMSE" for extent as it is exactly the Integrated Ice Edge Error (IIEE) of Goessling et al., (2016). You cite the ensemble extension of the IIEE (the SPS paper of Goessling and Jung, 2018) and say that your RMSE is like a deterministic version of that, which is misleading. It would be better to just cite the 2016 paper instead and call your metric "IIEE" instead of "RMSE": Goessling, H. F., Tietsche, S., Day, J. J., Hawkins, E., and Jung, T.: Predictability of the Arctic sea ice edge, Geophys. Res. Lett., 43, 1642–1650, https://doi.org/10.1002/2015GL067232, 2016*

Agreed - we have renamed the metric to IIEE to avoid confusion and replaced the reference with your suggested one.

*12. In Figure 3 I note that the neXtSIM concentration evolution is very smooth – more so than the low resolution SSMIS data – which I didn't expect given the resolution of the model. Can you comment on this? Is this caused by the fact that neXtSIM still uses the continuum formulation and so doesn't resolve small scale features?*

For this refer to p10 "For all comparisons we average the model fields in time over an appropriate time window (in practice 1, 2 or 7 days), apply some spatial smoothing (being guided by the size of the satellite footprint), and interpolate onto the observation grid." When comparing to OSISAF SSMIS we average over 1 day and apply some spatial smoothing, as SSMIS is too coarse to resolve the cracks/leads etc. However, we have applied a bit less smoothing this time around (see fig 11).

*13. I note with interest that MOSAiC forecasting is mentioned as a motivation for improving sea ice forecasts. There is an international project (SIDFEx) currently coordinating operational sea ice drift forecasts specifically to provide guidance to the Polarstern/MOSAiC. Presently the list of models includes TOPAZ but not neXtSIM. Are there any plans to contribute neXtSIM drift forecasts to SIDFEx? This might be an interesting way to show the skill of neXtSIM in this regard.*

This would be interesting if it were still possible, and is worth discussing with the leaders of that project.

*14. Some of the figures (e.g., Figs 10&11) suggest that the data assimilation is having a rather modest impact on the forecasts compared with many of the operational systems that I have seen in the past. Can you comment as to why that might be?*

After 2 quite major updates to our sea ice rheology we had to change the assimilation scheme quite a lot. It is now quite conservative in that we mostly leave the model alone, except where the ice mask is incorrect. There is a noticeable improvement in the IIEE for the first days after doing this, although the improvement is diminishing with lead time.

*15. On page 20 it is mentioned that the "RMSE for drifters placed on the first day. . ." but this is the 1st time in the manuscript that drifters are mentioned. Can you explain this a bit more please?*

Every day at 12:00 we place drifters at the grid points of the OSISAF drift product and advect them for 48h at the ice velocity. The total drift is then compared to the OSISAF drift product.

**Figures**

*As mentioned above I feel that there are too many figures in this manuscript. In particular in Figs 11 & 12 there are 12 panels and each row looks virtually identical. Apart from telling me that the assimilation is having a rather modest impact, I don't understand what I'm supposed to do with all this information.*

*Additionally the next similar set of figures, Figs 13-14, don't even seem to be discussed in the text at all. So are they necessary?*

We have revised the figures substantially now (eg figs 11-12 and figs 13-14 have been replaced.)

*Many of the figure captions are too brief and should be improved. I believe that the*

*Copernicus journal guidelines are that figures should be able to work stand-alone from the text, for which a bit  more information is required.*

We have now checked the captions contain enough information.

*I find that the x-axis date tick-marks provided on the time-series plots (Figs 3, 5, 7, 12 – less so for Fig 9) are not very useful. With such a short run period it would be better to include more dates. At the very least there minor tick marks should be used to show each day (or 5-days or something). It would also be good to specify this in the figure caption perhaps.*

We have tried to add enough major and minor tick marks to the time series plots, although the run is a lot longer now.

*Figure 1 is a bit confusing because I am left wondering whether different time-scales are involved here. Is this a daily schematic or does it depict the whole run? For example the 2 top boxes (initialization) are surely not done each day are they? If not then it should be made clear what is done each day and what isn't – either in the caption or the figure (or both). Perhaps the initialization boxes could be enclosed in a dotted box or something?*

*These were snapshots, but they have actually been removed to reduce the number of figures.*

*I suggest you should also re-think your use of red-blue colour maps for sea ice concentration. I have seen people use red for less ice (as it's hotter) and blue for more ice (colder) in the past as well as the other way around. It might be better to avoid the use of a hot<->cold colour-map therefore.*

*We tried out some other colormaps but decided to keep this one for lack of a better alternative.*

**Technical corrections**

*The 1st instance of "SSMIS" is correct but thereafter it has been changed to "SMMIS".*
Fixed

*Also "first day (4th day)" appears in many places, which is not very consistent*
*P2, L26: CMEMS should be "Copernicus Marine Environment Monitoring Service"*
Fixed

*P2, L29: ". . .the version 4.1 of the. . ." – suggest to remove the 1st instance of "the"*
Here
Fixed

*P2, L30: The reference for CICE v4.1 is Hunke and Lipscomb (2010): Hunke, E. C. and Lipscomb, W. H.: CICE: the Los Alamos sea ice model. Documentation and software users manual, Version 4.1 (LA-CC-06-012), T-3 Fluid Dynamics Group, Los Alamos National Laboratory, Los Alamos, 2010*
Thanks - added this

*P3, L30: "As specified by the validation reports above. . ." should be "As specified by the validation reports cited above. . ."*
Changed to this.

*P4, L5: "metrics" should be "metric"*
Fixed

*P4, L6: is extent "above 15%" or "at least 15%"? I thought the latter.*

In our evaluation script it was >15% - in practice I would expect either to give the same result

*P4, L13: ". . .can be obtained for 48 hours. . ." sounds like only 48 hours of data. Do you mean this or do you mean the data is available 48-hours behind real-time?*

In the OSISAF product, the drift is estimated daily from 12:00 on the start date until 48 hours later. In practice this does imply a time lag of at least 48h. "From October to April, however, daily 48-hour ice drift vectors can be obtained at a spatial resolution of 62.5 km."

*P5, L5: calculating volume for each model & obs based on thickness like this will involve different areas of ice won't it?*

Yes. In the end we decided to remove the SMOS comparison as errors were greatly affected by errors in extent (which we struggled with).

*P5, L26: "Modelling" is spelt incorrectly as "Modeling" in the NEMO acronym*

Fixed

*P5, L31: ". . .if the temperature is below 0C". Which temperature – surface skin temperature, or near-surface atmosphere temperature (T2M)? Please be more specific.*

It is T2M: we have added a clarification to the paper.

*P10, L29: "of these variables" is not adding to this sentence and should be removed*

Removed

*P11, L6: ". . .predicts the no ice. . .". Please remove the "the".*

Fixed

*P11, L11: "averages values" should be "average values"*

Fixed

*P11, L19: "underestimation in the Bering Sea". I presume that you mean the Chukchi Sea here because the Bering Sea is outside your model domain?*

*P13, L33-4: "land-fast" should be "land-fast ice"?*

Fixed

*P13, L35: (as above) I suspect that "Bering Sea" should be "Chukchi Sea" here*

That is correct, although with the change of simulation length the figures and discussion relating to the free run are quite changed and this sentence has been removed.

*P14, L9: the title "Evaluation of forecasts with assimilation" is confusing because I doubt that you are actually doing assimilation in your forecasts are you? Perhaps this should be changed to something more like "Forecasts performed from analysed ice conditions"?*

We are doing a kind of assimilation known as "data insertion" (modifying the initial conditions of the forecast) combined with "nudging" (through the compensating heat flux)

*P16, L9: Do you mean "significantly" here in the scientific sense of the word? If so include a p-value, if not I suggest changing to "considerably".*

Changed to 'considerably'

*P18, Fig 9 caption: "(blue)", "(orange)", and "(red)" are provided but not "(green)"*

This figure has now been removed.

*P21, Fig 12 caption: "error bars" should be "shading"*

We have removed references to error bars since we use shading.

*P26, L7: "limited resources" suggests a deficiency in resource. You should change to "minimal resources" if you wish to suggest that the model is cheap to run.*

We decided to remove this discussion.

*P26, L20-21: ". . .forecasts used saved atmospheric and ocean forecasts as forcing. . .".*
*What does this mean?*
This is mainly an issue of forecasts vs hindcast winds which we have clarified in a few
places (introduction, start of results section, discussion and conclusions)

---

## Referee Report (RR1)

2nd review of **Presentation and evaluation of the Arctic sea ice forecasting system neXtSIM-F** by Williams et al.

The authors have addressed my main concerns. I recognize the huge amount of work that was put in order to evaluate the performances of nextSIM-F and to improve the paper. I just have a few additional minor comments to improve the clarity of the text. I recommend that the manuscript could be published once these minor comments will have been addressed.

**1. Minor comments**

1) You often refer to viscoplastic models. To be consistent with the literature, I suggest you use the expression viscous-plastic.

2) p.1 line 12: ...greatly improveS...

3) p.3 line 5: Parenthesis are missing for the references.

4) p.2 line 22: OSISAF is not assimilated in RIPS. In the paper (Lemieux et al. 2016) it is written:

"Retrievals of sea ice concentration from passive microwave (Special Sensor Microwave Imager, SSM/I; Special Sensor Microwave Imager/Sounder, SSMIS) and advanced scatterometer data, and manually produced sea ice charts from the Canadian Ice Service (CIS) are assimilated by the 3D-Var system."

5) p.2 footnotes: The "S" in RIPS and RIOPS stands for System...not Service.

6) p.3 line 12: ...enterED into operations in...

7) p.3 line 23: I would be surprised that mariners currently plan their operations based on forecasts of leads. The only thing I heard of is that the US navy is using

NRL's lead forecasts for their submarines...

8) p.5 line 1: Please rephrase.

9) p.5 line 28: Please rephrase (I don't understand "at the ice velocity").

10) p.6 line 23: Rampal udated the paper of Rampal...this sounds weird. Something like: "Following the work of Rampal 2016, Rampal 2019 evaluated..."

11) p.6 line 24: "Addition" is repeated twice...please rephrase.

12) p.7 line 2: $k_1$ should be unitless while $k_2$ should be in $\mathrm{Nm}^{-3}$. Here is an important comment: The suggested (and optimized value) of $k_1$ is 8 not 10. This clearly explains why nextSIM overestimates landfast ice in the Laptev and East Siberian Seas. As nextSIM tends to simulate ice a bit too thick you could even use $k_1$=7.

13) p.8 section 3.2: It should be clearer that this is done only at the beginning of the free run and before the first forecast. Some people might be confused that this is done to initialize all the forecasts.

14) I am confused with all the different variables that are used for concentration ($c_t$, $c_U$, $c_y$, $c_F$, $c_B$, etc.). I am sure you can simplify this. For example, if I am right, $c_B$=$c_O$ and $c_t$=$c_F$.

15) p.9 line 19 (and at other places): It is not clear what you mean by ice mask.

16) p.10 line 26: You mean "lower" not "greater"?

17) p.11 line 4: Replace "reflected" by "reflecting"

18) Fig. 3: there is no shaded area as mentioned in the caption.

19) p.14 line 1: You refer to Jan-Feb in Figure 4 but it does not exist. Do you want to write "not shown"? Same idea for Nov-Dec.

20) Fig. 5: the text is very small. Please improve this.

21) p.22 line 13...: You use to much "we" in this paragraph. For example: "...we are systematically lower in concentration...". Replace by: "The forecasts are systematically...". Same idea for "...we score..."

Congratulations for your paper!

Jean-François Lemieux

---

## Author Response (AR3)

**Response to reviewer 1 (Jean-Francois Lemieux)**

**2nd review of Presentation and evaluation of the Arctic sea ice forecasting system neXtSIM-F by Williams et al.**

Thank you, Jean-Francois, for your review of our paper and your useful suggestions, that helped improve the paper a lot.

The authors have addressed my main concerns. I recognize the huge amount of work that was put in order to evaluate the performances of nextSIM-F and to improve the paper. I just have a few additional minor comments to improve the clarity of the text. I recommend that the manuscript could be published once these minor comments will have been addressed.

**1. Minor comments**

1) You often refer to viscoplastic models. To be consistent with the literature, I suggest you use the expression viscous-plastic.

We have changed this term.

2) p.1 line 12: ...greatly improveS…

Corrected.

3) p.3 line 5: Parenthesis are missing for the references.

Corrected.

4) p.2 line 22: OSISAF is not assimilated in RIPS. In the paper (Lemieux et al. 2016) it is written:

"Retrievals of sea ice concentration from passive microwave (Special Sensor Microwave Imager, SSM/I; Special Sensor Microwave Imager/Sounder, SSMIS) and advanced scatterometer data, and manually produced sea ice charts from the Canadian Ice Service (CIS) are assimilated by the 3D-Var system."

Corrected this description of RIPS.

5) p.2 footnotes: The "S" in RIPS and RIOPS stands for System...not Service.

Corrected.

6) p.3 line 12: ...enterED into operations in…

Corrected, since this date is now in the past.

7) p.3 line 23: I would be surprised that mariners currently plan their operations based on forecasts of leads. The only thing I heard of is that the US navy is usingNRL's lead forecasts for their submarines…

Changed "More pertinently in a forecast context, they are also highly relevant for navigation." to "While the precise forecast of individual leads is very challenging (and probably requiring assimilation of quite specific data like SAR-derived deformation, Korosov and Rampal, 2017), reliable information of this sort would be very useful for icebreakers that wish to reduce fuel consumption or submarines wishing to surface." (Thanks for the idea about submarines.)

8) p.5 line 1: Please rephrase.

Changed "In order to combine the advantages of these products we generated a blended product that was used both    for assimilation during the forecasts." to "In order to combine the advantages of these products we generated a blended product that was used for assimilation during the forecasts."

9) p.5 line 28: Please rephrase (I don't understand "at the ice velocity").

Clarified this procedure.

10) p.6 line 23: Rampal updated the paper of Rampal...this sounds weird. Something like: "Following the work of Rampal 2016, Rampal 2019 evaluated..."

Actually deleted this sentence.

11) p.6 line 24: "Addition" is repeated twice...please rephrase.

Rephrased.

12) p.7 line 2: $k_1$ should be unitless while $k_2$ should be in $Nm^{-3}$. Here is an important comment: The suggested (and optimized value) of $k_1$ is 8 not 10. This clearly explains why nextSIM overestimates landfast ice in the Laptev and East Siberian Seas. As nextSIM tends to simulate ice a bit too thick you could even use $k_1 = 7$.

Thanks - we have noted this comment. In the latest round of tuning k1=5 improved things a lot, and we will try k1=7 also.

13) p.8 section 3.2: It should be clearer that this is done only at the beginning of the free run and before the first forecast. Some people might be confused that this is done to initialize all the forecasts.

We have clarified this.

14) I am confused with all the different variables that are used for concentration ($c_t$, $c_U$, $c_y$, $c_F$, $c_B$, etc.). I am sure you can simplify this. For example, if I am right, $c_B = c_O$ and $c_t = c_F$.

We have endeavoured to simplify these variables this time.

15) p.9 line 19 (and at other places): It is not clear what you mean by ice mask.

We have added the definition the first time it occurs.

16) p.10 line 26: You mean "lower" not "greater"?

Yes - changed this sentence

17) p.11 line 4: Replace "reflected" by "reflecting"

Changed to 'reflecting'

18) Fig. 3: there is no shaded area as mentioned in the caption.

Corrected this caption

219) p.14 line 1: You refer to Jan-Feb in Figure 4 but it does not exist. Do you want to write "not shown"? Same idea for Nov-Dec.

This is correct - changed this sentence

20) Fig. 5: the text is very small. Please improve this.

Increased the font size for this figure

21) p.22 line 13...: You use to much "we" in this paragraph. For example: "...we are systematically lower in concentration...". Replace by: "The forecasts are systematically...". Same idea for "...we score..."

Changed to your suggestion

Congratulations for your paper!
Jean-François Lemieux
Thanks!

**Response to reviewer 2 (anonymous)**

Thank you, Reviewer 2, for your extremely thorough review of our paper and your useful suggestions, that helped improve the paper a lot.

**Overview**

I am pleased to see that the authors have addressed one of my main concerns from the last review by extending the hindcast runs, and associated assessment, out to more than 1 year. Given that things are worse in the second winter than the first, I feel it would be better to have run the system for even longer, but 20 months is much better than what we had before. Glad you feel this is better - we are constrained in going back further by the version of the atmospheric forcing product which began in the summer of 2018 (we initialise our forecast early in November of that year, as soon as the CS2-SMOS settles down).

However, I find myself still rather frustrated with the presentation of this manuscript. The wording used is still very informal/unscientific and the arguments are not laid out in a manner that is conducive to transfer information to the reader – i.e., detailing exactly what has been done, why certain decisions were made, etc.. Moreover, the figures are quite difficult to understand. Several of the newer figures use very small text and/or very small coloured dots, that make them almost illegible. Furthermore, nearly all of the figures have captions that are either wrong or don't contain the required detail, and all figures suffer from a lack of annotation/labelling or panel subtitling, that would allow the reader to quickly ascertain exactly what is plotted where.
We have endeavoured to improve the language, figures (particularly by increasing font size and adding some extra annotations) and captions.

Finally, after reading this through again, I'm afraid to say that I now find myself even more confused about exactly what is done in the neXtSIM-F operational forecasting system (and why!). I don't find Figure 1 very informative in this regard without the appropriate explanation. For example, I am not sure what the difference between the "initialization" and "assimilation" boxes – when most operational centres would actually class the assimilation as the initialisation (i.e., its only job is to get the forecast starting from the best possible initial conditions!). I am also left wondering why the assimilation seems to be being applied during the forecasts (P19 L29-31) and why it seems to make little difference either way. Given this last point, I am further left wondering what the analysis is about (i.e., what is analysed, how is it used, how does it differ from the 1-day forecasts that contain assimilation?).
I think this issue can be fairly easily fixed - the authors need to take some time to carefully explain the top-level structure of the forecasting system. I would recommend doing this right at the start of Section 3 where Fig 1 is introduced (i.e., before the model is discussed in Section 3.1).
We have endeavoured to improve the description and have also simplified Fig 1 to show only what happens every day.

Also I find that things are further confused by the dual discussion of MEB and BBM versions of neXtSIM but only the latter is evaluated here. In some places (e.g. Appendix A1) this feels like a CMEMS report about the operational system (in particular given that December 2020

is in the past)! Given that in this paper you are solely showing results from the BBM version of neXtSIM, much of the MEB stuff is not relevant - except to show the differences in the new (current?) system you are documenting here. So I feel it would make sense to tone the MEB descriptions down and just to focus on the new/current operational system.
This discussion/description is now much reduced.

**General comments**

You need to be careful when talking about other systems all using the VP rheology because many readers will likely ask "what about EVP?". Many will not know how similar these are so it might be best to elaborate a little and say "based upon the viscous-plastic (E)VP formulation" or "the VP family of rheologies" or something. Also some might ask "what about the EAP" rheology? I think that RASM are using EAP for their coupled seasonal forecasts now(?). Maybe you could avoid that potential conflict by better quantifying the area you are talking about by stating "short-range analysis and forecasting system" instead of just "forecasting system"?
We have mentioned the EVP now as a method of solving the VP equations
**NB. page and line numbers used below correspond to those used in the tracked-changes version of the document. I now notice that these are somewhat odd with some pages having some line repetition**

**Specific comments**

P2 L15: "while Hunke et al. (2020, Table 1) give a comprehensive list of modelling systems that include sea ice": I would be a little careful here because the Hunke et al. paper is actually about the use of climate model sea ice modelling formulations for operational forecasting. Therefore, it does not aim to give an overview of all operational forecasting systems (certainly not all "modelling systems" as you claim!), instead only those using the classic AIDJEX continuum model formulation (CICE, LIM/SI3, MITgcm).
While this paper was not explicitly aiming to give an overview of forecasting systems, it did nevertheless give a long list of them in table 1 which made it a convenient paper to refer to. We changed this sentence to:
"Tonani et al. (2015) give a good overview of the 2015 status of operational forecasting (here we take "operational forecasts" to refer to those with forecast horizons of about a week), while Hunke et al. (2020, Table 1) give many examples of modelling systems that include sea ice, most of which are used operationally in national forecasting capacities."

P2 L18: "[models/systems] do not vary in their numerical framework": I would change the wording here because many would argue that there are lots of differences between models/systems - with some models having complicated thermodynamics schemes and others not having sea ice thermodynamics at all (zero-layer). If you mean the dynamics and the overall structural formulation (i.e., AIDJEX continuum dynamic-thermodynamic models) then you should say so.
Good point - we have clarified that we are referring to the sea ice dynamics. The sentence is now: "We note however that their sea-ice dynamics schemes are all based on Eulerian advection schemes and on variants of the viscous-plastic (VP) rheology (although some solve the rheological equations directly while others solve modified equations as is done with the elasto-viscous plastic method (EVP))."

P2 L24: I don't quite get the RIPS->RIOPS paragraph here (aside from J-F requesting it!) because it almost invalidates your argument for running standalone neXtSIM. You need more discussion here, I think. For instance, why did ECCC move away from the basic/standalone RIPS forecast towards RIOPS? What physics/skill does neXtSIM-F likely miss out on by sticking with the "RIPS" style approach (i.e., standalone) rather than the full "RIOPS"?

Good point. It seems like one of the main motivations for RIOPS was having forecasts of currents and tides for search and rescue reasons. There are some drawbacks to the stand-alone approach of course (which is our only possible approach currently), and we added some text, giving the ice edge location as an example.

P3 L8-10: again with the first forecasting system not to use VP you should be careful of RASM using EAP and to ensure that EVP is captured (as in General comments)

We have now clarified VP/EVP and that we are not dealing with seasonal forecasting earlier in the paper.

P4 L20: ocean forcing – please specify the frequency of forcing field update – daily? Hourly?

Daily - changed the text.

P4 L20: "TOPAZ near-surface (30m) velocity": Is this exactly the velocity at 30m or the integrated velocity over the top 30m? You need to be more specific. Either way can you say you use this approach?

This was actually a mistake - we currently use the surface (0-3m) currents. Changed the text.

P5 L10: atmos forcing – please specify the frequency of forcing field update – daily? Hourly?

6-hourly

P5 L25: "(European western time)": do you mean UTC? If so that would be easier to understand.

Actually it is Central European Time (CET) since our server runs on Bergen time. Fixed this in the text.

P6 L12-3: "(It is therefore an independent validation dataset for our forecasts.)": I find this odd because the previous sentence – i.e., used for evaluation of free run and forecasts – does not make the data independent! Needs rewording.

Changed to: "It is not assimilated."

P6 L19: "As part of our evaluation we sometimes apply a filter on the uncertainty…": I don't understand why we have sometimes here! Did you do this filtering here or not? If so remove the sometimes; if not why mention it?

Rephrased this section (we always do it but sometimes vary the error threshold for retaining an observation or not).

P7 L11: "In order to compare neXtSIM drift…": You don't need to do this to compare drift though do you? You could use the model velocity fields? The point here is that you are trying to make a better comparison and "compare apples with apples" so why not say that?

Clarified this procedure, to explain better that in this way the drift can be updated every model time step as opposed to every hour (time resolution of the CMEMS product)

P7 L13: "…place Lagrangian drifters…": I would be more explicit and say that you "…seed synthetic Lagrangian drifters into the model…".

We changed this phrase to your suggestion.

P7 L15: Sentence starts with "We use this product" but you've not defined/identified a product yet! Unless you include the "CS2-SMOS" in the subtitle (which I don't). Needs rewording.

Fixed

P7 L27-28: "A delay of one week for thickness would probably be acceptable for assimilation in a real time forecast in the future." You need to be careful here because many others (including me) would be of the opposite opinion. I guess you mean that you can pretend that the thickness from 7 days ago is the thickness from today and assimilate it? Even so that doesn't sound ideal. If however you're talking about weekly thick ice from CS2 but still using daily SMOS data for thinner ice then perhaps that is ok. However you don't say these things! Either way I would reword to make things clearer. Many of us are pushing hard to get satellite SIT available for near-real-time assimilation and so any statements along the line of what you say here might derail that. So you need to be clear and careful.

Good point - we are certainly thinking about the thicker ice and would have a fairly slow nudging (which also introduces an additional time lag) in order to stop the thicker ice drifting too much from observations, and also recognising there are a lot of uncertainties in these observations. We just deleted this sentence.

P11 L4: "The mesh is generated with Unref (a component of Gmsh…" needs more information to explain what these are. Are they numerical packages or publicly available tools or something?

Changed to "Unref (a component of the open-source mesh-generation library GMSH: Geuzaine and Remacle, 2009)"

P11 L26-29: regarding satellite observations being interpolated to model grid - This is the opposite way around that most people do assimilation, and model-observations comparisons generally, where the model is always translated to the observational locations. Why do you do this the other way around? What impact might this have?

It is done this way mainly to keep the assimilation as simple as possible, especially given the relatively unsophisticated system (e.g. without an ensemble to determine model covariances). However, we agree that this is not the usual way of comparing model to observations. There could be some benefit to interpolating to the observation grid (as you suggest), smoothing the model to allow for a larger satellite footprint and then using this comparison to do the update. It is hard to know exactly how much difference this would make without implementing it, but it is not a priority.

P12 L17: "(this is a kind of assimilation of extent)" given that you don't use the concentration in the pack why do you even need to do the assimilation given that the ice edge would come through in the SMOS SIT initialisation anyhow?

We have now tried to distinguish better in the paper between initialisation of the forecast (done once only, using CS2-SMOS) and the daily assimilation (only uses OSISAF concentration). Thus we don't assimilate SMOS SIT daily. There could be some benefit to adding assimilation of SMOS in winter to constrain the thickness of new ice. However we note the ice edge from SMOS is very low resolution.

P12 L3 (bottom of page!): "the heat flux out of the ocean increases and the ice freezes up again very fast". Surely this is not always the case? Only if the atmospheric forcing thinks there should be ice? i.e., if there is ice in the ECMWF model used to provide the then the atmospheric fields will be cold and conducive to ice regrowth but if, however, there is no ice in the ECMWF model then the near-surface atmosphere will be warmer?

We found the most dramatic refreezing of removed ice to be due to periods of very cold atmosphere. Rephrased this explanation.

P15 L4: Re Figs 2&3 - what is "mean concentration"? Is this the mean over the whole domain, only ocean points, only sea ice point, or what?

Mean over ocean points.

P17 L12-14: both "Jan-Feb" and "Nov-Dec" are discussed here in relation to Fig 4 but neither panels are present in Fig 4?!

Rephrased this discussion since they are not shown.

P24 L4 (bottom page): "…with 1-day forecasts being launched in between so assimilation still performed daily." Are these not the analyses?

Yes.

P25 L11: (equation 7): why, when you say IIEE is analogous to RMSE, is the RMSE ratio squared for concentration (eq7a) but the IIEE ratio is not squared for extent (eq7b)?

Rephrased this sentence - we meant that it is like the RMSE in that it is also a positive-definite error metric.

P25 L12-15 (bottom page): this needs rewording. I have no idea what "straw man" means here – are you seeking feedback on a basic idea from others? or trying to scare birds away from your crops? Neither seem appropriate. Also "We have rough benchmarks from some other models however" doesn't read well. If this relates to the next sentences then you might as well delete it. If not, what are these benchmarks?

A straw man is an expression for an easy-to-beat opponent, but this is probably too colloquial, so it was changed. We rephrased these sentences. The "benchmark" sentence now reads "However, we can make some rough comparisons to the drift errors of other products, even though they use different observations for their evaluation so we can't make a direct comparison. We note that the drift from the TOPAZ forecast generally has a bias in speed of about 2 km/day and a VRMSE of about 5 - 8 km/day (Melsom et al, 2018), while Metzger et al ( 2017) report an RMS drift speed error of about 5 - 8 km/day in the Arctic for the GOFS 3.1 system."

P27 L12-13: "we don't try to correct this as reducing the concentration in the pack causes serious problem with the drift and thickness": this might work well for a standalone sea ice model but you could have trouble with heat fluxes if you coupled to the ocean and/or atmosphere. Can you say any more about this – particularly in light of the last line of the paper where you mention plans to couple with ocean and/or waves.

We agree that such a method could cause problems when coupled to an ocean or atmospheric model. The ongoing work on using the ENKF assimilation could possibly help - if the model always has 100% concentration in the pack (at least in the winter) then probably the model variance will be very low there compared to the observation error, and the correction would probably be quite small - possibly small enough not to cause problems for the drift and thickness. Another approach could be to let the model variance drop to zero in a smooth way as distance from the edge increases. Another solution could be to use a different product which had a higher concentration in the pack.

**Figures**

Figs 2 & 3: the text used for legend & axes is very small and hard to read comfortably

Increased the font size for these figures

Fig 4 (caption): the caption should state at the start that these are "differences" between the free run and OSI-SAF

Fixed figure caption

Fig 5: the text is far too small in this figure and the caption makes no mention of what the left and right panels show (I had to zoom in to 200% to see the dates to work out they are the 2 winters - 18-19 and 19-20)!

Increased the font size for this figure and added titles to the 2 columns ("2018-2019 winter", "2019-2020")

Figs 6 & 8: panel titles or annotation would make it much easier to see what is plotted in each column

Added annotations

Figs 9 & 12: The detail on this figure is tiny. Even at 200% zoom I struggle to see what's going on properly - particularly the tiny coloured dots.

Increased the font size for these figures

Fig 9 (caption): the caption is all wrong and relates to 3 rows when there are only 2.

Fixed caption

Figs 10 & 11: what do the columns show? panel titles or annotation would make it much easier to see what is plotted in each column

Added annotations

**Typos**

P1 L4 (and other locations): VP should be "viscous-plastic" not "viscoplastic"

Fixed

P2 L12: "MOSAIC" should be "MOSAiC"

Fixed

P2 L21; "operation" should be "operational"?

Fixed

P3 L11: CMEMS = "Copernicus Marine Environment Monitoring Service". Your version with "Marine and Environmental" would be a very different beast rather than just doing the "Marine Environment"!

Fixed

P4 L9: Arctic MFC = "Arctic Monitoring and Forecasting Centre" (see https://marine.copernicus.eu/about/producers/arctic-mfc)

Fixed

P4 L12: (as mentioned in the last review) the reference for CICEv4.1 is "Hunke and Lipscomb, (2010)" not "Hunke et al.". It looks like in the references you have changed the date to 2010 but not changed any other details of the reference - including the author list! "Hunke, E. C. and Lipscomb, W. H.: CICE: the Los Alamos sea ice model. Documentation and software users manual, Version 4.1 (LA-CC-06-012)."

Fixed

P4 L30: You should drop the "both" from "used both for assimilation" now that "and evaluation" is deleted

Fixed

P5 L19: AMSR2 = "Advanced Microwave Scanning Radiometer 2"

Fixed

P6 L17: "Cryosat-2" should be "CryoSat-2"

Fixed